# Timosaponin A3 Inhibits Palmitate and Stearate through Suppression of SREBP-1 in Pancreatic Cancer

**DOI:** 10.3390/pharmaceutics14050945

**Published:** 2022-04-27

**Authors:** Yumi Kim, Wona Jee, Eun-Jin An, Hyun Min Ko, Ji Hoon Jung, Yun-Cheol Na, Hyeung-Jin Jang

**Affiliations:** 1College of Korean Medicine, Kyung Hee University, 26 Kyungheedae-ro, Dongdaemun-gu, Seoul 02447, Korea; yumi0201@khu.ac.kr (Y.K.); 97wona@naver.com (W.J.); aej3866@naver.com (E.-J.A.); rhgusals93@naver.com (H.M.K.); johnsperfume@gmail.com (J.H.J.); 2Department of Science in Korean Medicine, Graduate School, Kyung Hee University, 26 Kyungheedae-ro, Dongdaemun-gu, Seoul 02447, Korea; 3Western Seoul Center, Korea Basic Science Institute, 150 Bugahyeon-ro, Seodaemun-gu, Seoul 03759, Korea; nyc@kbsi.re.kr

**Keywords:** SREBP-1, pancreatic cancer, timosaponin A3, fatty acids, tumor growth

## Abstract

Timosaponin A3 (TA3) was demonstrated as a potent anticancer chemical by several studies. Although the effects of inhibiting growth, metastasis, and angiogenesis in various cancer cells were demonstrated through multiple mechanisms, the pharmacological mechanism of TA3 shown in pancreatic cancer (PC) is insufficient compared to other cancers. In this study, we aimed to explore the key molecular mechanisms underlying the growth inhibitory effects of TA3 using PC cells and a xenograft model. First, from the microarray results, we found that TA3 regulated *INSIG-1* and *HMGCR* in BxPC-3 cells. Furthermore, we showed that inhibition of sterol regulatory element-binding protein-1 (SREBP-1) by TA3 reduced the fatty acid synthases FASN and ACC, thereby controlling the growth of BxPC-3 cells. We also tried to find mechanisms involved with SREBP-1, such as Akt, Gsk3β, mTOR, and AMPK, but these were not related to SREBP-1 inhibition by TA3. In the BxPC-3 xenograft model, the TA3 group had more reduced tumor formation and lower toxicity than the gemcitabine group. Interestingly, the level of the fatty acid metabolites palmitate and stearate were significantly reduced in the tumor tissue in the TA3 group. Overall, our study demonstrated that SREBP-1 was a key transcription factor involved in pancreatic cancer growth and it remained a precursor form due to TA3, reducing the adipogenesis and growth in BxPC-3 cells. Our results improve our understanding of novel mechanisms of TA3 for the regulation of lipogenesis and provide a new approach to the prevention and treatment of PC.

## 1. Introduction

Pancreatic cancer (PC) has an extremely poor prognosis and a high mortality rate [1]. Despite the advances in PC management, the five-year survival rate was a dismal 11% in the United States from 2011 to 2017, largely due to the challenges in treating the disease owing to drug resistance and metastases [2]. Symptoms of pancreatic adenocarcinoma (PA), which accounts for approximately 85% of pancreatic cancer cases, do not appear in the early stages of the disease [3]. According to the National Comprehensive Cancer Network (NCCN) guidelines for pancreatic adenocarcinoma [4], adjuvant gemcitabine monotherapy or adjuvant 5-fluorouracil (5-FU) monotherapy has been used for a long time for improvement after surgical resection. Moreover, adjuvant FOLFIRINOX (oxaliplatin, leucovorin, irinotecan, and 5-FU) chemotherapy or a combination of gemcitabine and capecitabine should be considered for patients with a good performance status [5]. However, even these drugs rarely prolong median survival by more than 6 months compared to people not taking the drug [6]. Several studies suggested a strong correlation between phytochemicals and reduced incidence of PC [7]. Phytochemicals present in plants are a promising option to increase treatment efficacy and reduce side effects in cancer patients.

Timosaponin A3 (TA3), which is a natural steroidal saponin isolated from *Anemarrhena asphodeloides* Bunge, has multiple pharmacological activities [8]. TA3 affects several cellular signaling pathways and shows efficacy in different cell types and disease models both in vitro and in vivo, including cancer, Alzheimer’s disease, depression, diabetes mellitus, and colitis [8,9]. TA3 markedly inhibited breast and hepatocellular cancer cell growth at the micromole level and selectively reduced the viability of cancer cells, but not that of normal cells [8,10]. The following papers demonstrated the potential of TA3 as a therapeutic agent for cancer. However, only three studies have investigated its role in PC thus far, including our previous study [11,12,13]. These studies demonstrated that TA3-mediated effects included apoptosis, anti-proliferation, anti-angiogenesis, and anti-proliferation by inhibition of PI3K/AKT, STAT3, and ERK. However, the molecular mechanisms underlying the anticancer effect of TA3 on PC cells and xenograft models remain largely unclear.

Many enzymes involved in the fatty acid synthesis and cholesterol synthesis pathways are regulated by sterol regulatory element-binding proteins (SREBPs) [14]. An SREBP, as a transcription factor, is bound by insulin-induced gene 1 (INSIG-1) and the SREBP cleavage-activating protein (SCAP)/SREBP complex remains an inactive precursor in the ER (endoplasmic reticulum). SREBP is activated by proteolytic cleavage of the Golgi apparatus to produce an N-terminal mature transcription factor that translocates to the nucleus [15]. It has two SREBP genes, namely, SREBP-1 and SREBP-2. SREBP-1 regulates enzymes involved in fatty acid synthesis, such as acetyl-CoA carboxylase (ACC) and fatty acid synthase (FASN) [16]. SREBP-2 is involved in the cholesterol synthesis pathway and activates the expression of genes such as HMG-CoA reductase (HMGCR), HMG-CoA synthase (HMGCS), and LDL receptor (LDLR) [17]. Over the past decade, lipogenesis was shown to be important in cancer growth, where it is highly linked to both glucose and glutamine metabolism [18]. Thus, recent SREBP-1 studies have highlighted a potential target for anti-cancer therapy that leads to the inhibition of proliferation [14,19]. Sunami reported that SREBP-1 levels were significantly higher in pancreatic cancer than in normal tissue adjacent to the tumor for 60 pancreatic cancer patients. Moreover, a high expression of SREBP1 indicated an association with clinicopathological features, such as tumor differentiation, a tumor-node-metastasis stage, and lymph node metastasis [20]. In this study, for the first time, we investigated the large-scale gene expression underlying the anticancer effects of TA3 in a human PC cell line, namely, BxPC-3, using microarray analysis. After the genomic analysis, we performed Western blotting and quantitative real-time-polymerase chain reaction (qRT-PCR) to validate the microarray results. Furthermore, we evaluated the anticancer efficacy of TA3 using the BxPC-3 xenografted in an immunodeficient mice model to further confirm the regulatory mechanisms in vivo. Finally, our study suggested the key molecular mechanisms underlying the growth inhibitory effects of TA3 in PC cells and a xenograft model.

## 2. Materials and Methods

### 2.1. Chemicals and Reagents

TA3 was purchased from Wuhan Chem Faces Biochemical Co., Ltd. (Wuhan, China). RNA isolation kits and cDNA synthesis kits were purchased from GeneAll Bioscience (Seoul, Korea). The SYBR green master mix was purchased from Life Technologies (Carlsbad, CA, USA). The lysis buffer and cell fractionation kit were purchased from Cell Signaling Technology (Beverly, MA, USA). Antibodies for SREBP-1 (SC-13551), p21 (SC-817), caspase-3 (SC-7272), p27 (SC-528), PARP (SC-7150), β-actin (SC-47778), and secondary antibodies (anti-rabbit, anti-mouse) were purchased from Santa Cruz Biotechnology (Santa Cruz, CA, USA) and the other antibodies were purchased from Cell Signaling Technology (Beverly, MA, USA). These antibodies were as follows: ACC (#3676), FASN (#3180), AMPK (#2532), phospho-AMPK (#2535), Akt (#4691), phospho-Akt (#4060), GSK3β (#9315), phospho-GSK3β (#9322), mTOR (#2972), phospho-mTOR (#2971), and lamin B1 (#13435). Dexamethasone (DEX), insulin, IBMX, heptadecanoic acid, palmitic acid, and stearic acid were purchased from Sigma-Aldrich (St. Louis, MO, USA). The ImmPRESSTM reagent kit was purchased from Vector Laboratories (Burlingame, CA, USA). Enhanced chemiluminescence (ECL) solution was obtained from DOGEN (Seoul, Korea).

### 2.2. Cell Culture and Treatment

Human PC cell lines (AsPC-1, BxPC-3, and PANC-1) were obtained from the American Type Culture Collection (Manassas, VA, USA). The cell lines were cultured in RPMI 1640 medium containing 10% FBS, 100 units/mL penicillin, and 100 µg/mL streptomycin at 37 °C in a water-saturated atmosphere with 5% CO_2_. For the experiments, the cells were treated with various concentrations of TA3 for 12 and 24 h. For the insulin studies, cells were pretreated with 1 µg/mL insulin for 30 min and then treated with TA3 for 12 and 24 h.

### 2.3. Cell Viability Assay

Cytotoxicity of TA3 on PC cell lines and 3T3-L1 cells was determined using the MTT assay. Cells were cultured in 96-well plates (1 × 10^4^ cells/well) overnight and then treated with different concentrations of TA3 for 24 h. MTT solution (2 mg/mL) was added to each well to a final concentration of 0.5 mg/mL and incubated for 4 h in a CO_2_ incubator. The MTT solution was discarded and the formazan crystals were dissolved in DMSO. The optical density was measured at 450 nm using a Multiskan™ GO Microplate spectrophotometer (Thermo Fisher Scientific, Vantaa, Finland).

### 2.4. Microarray Analysis

After the TA3 (5 μM) treatment, the cells were lysed using a RiboEX reagent. Total RNA was extracted using an RNA isolation kit according to the manufacturer’s protocol. For the microarray analysis, total RNA samples (500 ng) were amplified to generate amplified RNA (aRNA). Labeled aRNA was hybridized to an Affymetrix Human Gene 2.0ST array using the GeneChip hybridization system. After hybridization, the chips were washed and stained with a Fluidics station 450. Data analysis was performed using a GeneChip scanner 3000DX (AFFYMETRIX, INC., Central Expressway Santa Clara, CA, USA). Gene network analysis was performed using the online resource GeneMania to investigate mechanisms of high association or lists of interacting gene networks from microarray-obtained gene lists [21].

### 2.5. Oil Red O (ORO) Assay

Both cell lines (1 × 10^6^ cells/well) were incubated with TA3 (2.5 and 5 µM) for 48 h in 6-well plates. The cells were washed and fixed in 4% paraformaldehyde for 1 h and then stained with pre-warmed 0.25% ORO working solution for 15 min at 60 °C in an oven. After washing twice with PBS, the cells were photographed under a light microscope at magnifications of 100× and 400×. To quantify the range of adipose conversion, 1 mL of isopropanol was added to the plate and the optical density was measured at 500 nm using a Multiskan™ GO Microplate spectrophotometer (Thermo Fisher Scientific, Vantaa, Finland).

### 2.6. Immunocytochemistry for SREBP-1 Localization

To evaluate the SREBP-1 localization with or without TA3 treatment, cells were washed three times with PBS and fixed with 4% paraformaldehyde for 15 min at room temperature (RT). The cells were permeabilized with 0.2% Triton X-100 for 20 min, washed three times, and then blocked with 5% bovine serum albumin for 1 h at RT. They were then incubated overnight at 4 °C with anti-phospho-SREBP-1 (1:100) antibodies, washed three times, and incubated with Alexa Fluor 555 donkey anti-mouse antibodies for 1 h at RT. Next, the cells were stained with 1 µg/mL DAPI solution and mounted on glass slides using a fluorescent mounting medium. Fluorescence imaging was performed using the CELENA^®^S digital imaging system (Anyang-si, Gyeonggi-do, Korea), and DAPI and Alexa Fluor 555 stains were excited (Ex: 358 nm and 555 nm) and detected (Em: 461 nm and 580 nm).

### 2.7. Cell Lysis and Immunoblotting

Total cell lysates were prepared by lysing the cells using a lysis buffer. The cell lysates were fractionated into cytoplasmic and nuclear components using a cell fractionation kit. After lysis and fractionation, the total protein concentration was determined using the Bradford reagent and immunoblotting was performed as described previously [22]. Briefly, equal amounts of proteins were loaded and separated using 8% or 15% sodium dodecyl sulfate-polyacrylamide gel electrophoresis (SDS-PAGE) gels and transferred to nitrocellulose membranes. The blots were incubated with primary antibodies (1:1000) overnight, followed by incubation with HRP-conjugated secondary antibodies (1:8000) for 1 h and visualization with enhanced chemiluminescence solution using ImageQuantTM LAS500 chemiluminescence (GE Healthcare Bio-Sciences, Uppsala, Sweden). The bands were quantified using Image J software. Fold change was calculated using the normalized values. The phosphorylation form was normalized to the levels of the total form and single proteins were normalized using loading control proteins (β-actin or lamin B1).

### 2.8. Cell Cycle Analysis

Based on a previous study [11], BxPC-3 cells (5 × 10^5^ cells/well) were plated in a 6-well plate and treated with TA3 for 24 h for cell cycle analysis. Cell cycle analysis was performed using propidium iodide (PI) staining. FACScan Calibur flow cytometry (BD Biosciences, Franklin Lakes, NJ, USA) was used to analyze the DNA content in the PI stained cells. Data acquisition and analysis were performed using Cell Quest 3.0f software (BD Biosciences).

### 2.9. Apoptotic Assay

Apoptosis was detected using an Annexin V/FITC apoptosis detection kit (BD Biosciences) following the manufacturer’s instructions. After treatment, the cells were stained with Annexin V/FITC or PI (1 μg/mL). They were then analyzed using a FACScan Calibur flow cytometer (BD Biosciences). Data were analyzed using Cell Quest 3.0f software (BD Biosciences).

### 2.10. Quantitative Real Time-PCR

After treatment, total RNA was extracted from the cells using an RNA isolation kit. cDNA synthesis was performed according to the kit protocol, and qRT-PCR analysis was performed as described previously [11]. Briefly, PCR amplifications were always performed in duplicate wells following the SYBR Green PCR Master Mix manufacturer’s thermal cycling parameters: 95 °C for 10 min as a hold step, then 40 cycles were performed using two steps per cycle (95 °C for 15 s as a denaturing step and 60 °C for 60 s as an annealing step). The relative expression of genes was analyzed by PCR using GAPDH as an internal control. The primer sets used for PCR are listed in Table 1.

### 2.11. siRNA Transfections

Human SREBF1 siRNAs (6720, Bioneer, Daedeok-gu, Daejeon, Korea) and control siRNAs (Bioneer) were transiently transfected into cells with Lipofectamin 2000 transfection reagent according to the manufacturer’s instructions (Thermo Fisher Scientific, Carlsbad, CA, USA). After 24 h, the cells were harvested for immunoblotting.

### 2.12. Animals

All procedures involving animals were reviewed and approved by the Institutional Animal Care and Use Committee of the Kyung Hee University (KHUASP(SE)-18-175) (14 February 2019). Five-week-old athymic nu/nu female mice (NARA Biotech, Seoul, Korea) were subcutaneously injected with BxPC-3 cells in the right flank. Animals were housed (6 mice/cage) in standard mice Plexiglas cages. Animals were fed normally sterilized mouse chow with water ad libitum and were acclimatized for a week before the experiment in a room maintained at constant temperature and humidity under a 12 h light–dark cycle. As a result of monitoring during the adaptation period, all mice were pathogen-free and healthy, showing no lesions.

### 2.13. Tumor Xenograft Experiments

Collected BxPC-3 cells were transferred to a 50 mL conical tubes and rinsed once in a serum-free medium. Next, cells were resuspended in PBS and calculated to provide 1 × 10^7^ cells per mouse. For the injection, 100 μL of BxPC-3 cells with PBS and 100 μL of Matrigel were mixed on ice before being injected subcutaneously into the right flank of all the mice. After 7 days, we randomized six mice to each group and intraperitoneally (i.p.) injected them with 100 μL of PBS (vehicle group) or 5 mg/kg body weight TA3 (TA3 group) and gemcitabine (GEM group) in the vehicle- and drug-treated groups thrice a week. Each drug treatment was performed for 20 days with a volume measurement of the tumor; 5 days after the end of treatment, euthanasia was performed to excise the primary tumor, and the final tumor volume was measured as V = 4/3 πr^3^, where r is the mean radius of the three dimensions (length, width, and depth). Half of the tumor tissue was fixed in formalin, while the rest was immersed in liquid nitrogen to be rapidly cooled and stored at −80 °C.

### 2.14. IHC

After dissection, formalin-fixed tumor tissues were rinsed in PBS, processed, and embedded in paraffin according to our previous IHC protocol. Tissue sections of 5 µm thickness were deparaffinized in xylene, rehydrated in graded ethanol, and finally hydrated in water. Antigens were retrieved by boiling the slides in 10 mM sodium citrate (pH 6.0) for 30 min. IHC was performed using the ImmPRESSTM reagent kit according to the manufacturer’s instructions. According to the instructions, the slides were incubated with the blocking reagent to block non-specific binding. Sections were then incubated overnight with primary antibodies against PARP, cleaved caspase-3, p21, p27, SREBP-1, and FASN (1:100 dilution). After the slides were washed, the slides were incubated with 3,3-diaminobenzidine tetrahydrochloride (DAB) for 1 min to detect immunoreactive sites (stained brown). Then, the slides were stained with Gill’s hematoxylin and mounted on glass coverslips using the mounting solution. Images were captured using an eXcope T500 microscope camera (magnification, 400×). The area fraction of each DAB stain was calculated using the ImageJ IHCtool Box. Statistical analysis was performed using GraphPad Prism software (GraphPad Prism, San Diego, CA, USA).

### 2.15. Preparation of Sample and Fatty Acid Standards

Frozen tumor tissue was ground into a fine powder in liquid nitrogen using a mortar and pestle. Tumor tissues were prepared using a chloroform–methanol–water extraction as per the Folch method [23]. Five milligrams of tumor tissue were homogenized with 1 mL of chloroform–methanol (2:1, *v/v*) mixture. The homogenate was mixed with 0.2 mL of 1 mol/L sodium chloride and centrifuged at 3000 rpm for 5 min. The bottom chloroform layer containing lipids was evaporated using nitrogen and redissolved in 100 µL of methanol. Calibration standards were prepared in 100% methanol at concentrations of 0.5, 0.75, 1, 2.5, 5, and 7.5 mg/L for palmitic acid and 0.1, 0.25, 1, 2.5, 5, and 7.5 mg/L for stearic acid. Internal standards were added to all calibration standards and samples at 1 mg/L for heptadecanoic acid.

### 2.16. Quantitative Analysis of Fatty Acids in Tumor Tissue Using UPLC/Q-TOF-MS

After the preparation of the samples and fatty acid standards (described in the Appendix A), chromatographic separation of the tumor tissues was performed with an Agilent 1290 Infinity LC (Agilent Technologies, Walbronn, Germany) using an Acquity UPLC BEH C8 column (50 × 2.1 mm, 1.7 µm particle size, Waters) at 40 °C. Gradient elution of solvent A was water–methanol (97:3, *v/v*) with 10 mM ammonium acetate and 15 mM acetic acid (pH 4) and solvent B was 100% methanol. The gradient program was set to the following: 5–50% B (0–5 min), 50–90% B (5–17 min), 90% B (17–20 min), and the column was then equilibrated with 5% B for 3 min at a flow rate of 0.3 mL/min. All samples and calibration standards (1 µL each) were injected into the column using a Thermostated HiP-ALS autosampler. The HPLC system was interfaced with the MS system, which was an Agilent 6550 Q-TOF (Agilent Technologies, Walbronn, Germany) equipped with a jet stream ESI source operating in negative ion mode. The ESI spray voltage was set to 3500 V (Vcap). Mass spectra were acquired at a scan rate of 1.0 spectra/s with a mass range of 100–1200 *m/z*. The procedure was controlled using Q-TOF Quantitative Analysis software (Version B 07.00, Agilent Technologies, Walbronn, Germany).

### 2.17. Statistical Analysis

All data represent at least two separate experiments performed in triplicate. The significance of the data was analyzed using Graph Pad Prism Version 6.0, (Graph Pad Software Inc., San Diego, CA, USA) with a Mann–Whitney U test. In the graphs shown, the bars represent the mean ± SD. A *p*-value < 0.05 was considered statistically significant.

## 3. Results

### 3.1. TA3 Regulated Lipid Metabolism in Pancreatic Cancer Cells

Prior to performing the microarray analysis, we compared the cytotoxicity of TA3 in three PC cell lines (AsPC-1, BxPC-3, and PANC-1) using the MTT assay to select one cell line for the microarray analysis. As shown in Figure 1a, 10 µM of TA3 reduced the viability to less than 50% compared to controls in all three cell lines. In particular, we found that the BxPC-3 cells showed induced cytotoxicity, even at a low concentration of TA3 (1.25 µM). Therefore, we performed microarray analyses of BxPC-3 cells treated with TA3 (5 µM) for 24 h, and the comparative results are shown in Figure 1b and Table 2. The Figure 1b results show that 489 genes were differentially expressed with a 1.5-fold change in expression. A total of 158 genes were upregulated and 331 genes were downregulated in the TA3 group compared to the control group. The downregulated genes were involved in the cell division, DNA replication, G1/S transition of the mitotic cell cycle, and telomere categories. The upregulated genes were involved in the lipid metabolism, cholesterol metabolism, oxidation–reduction processes, and cell adhesion categories. In particular, 158 genes were related to cell cycle function, accounting for a large proportion (32.3%) of the total genes, with a 1.5-fold change in expression. The number of genes showing changes in expression related to the lipid metabolism process accounted for a small proportion of 5.9% (out of 489 genes). As shown in Table 2, the 22 genes were classified in terms of oncological function and over a 1.5-fold-change after TA3 treatment. The most altered gene expressions after TA3 treatment were hydroxymethylglutaryl-CoA synthase-1 (HMGCS-1; 4.51-fold change) and insulin-induced gene-1 (INSIG-1; 4.12-fold change) compared to the control group. To confirm whether TA3 (5 μM) increased the HMGCS-1 and INSIG-1 levels as same with microarray results, extracted RNA was analyzed at the mRNA level of HMGCS-1 and INSIG-1 from each of the three PC cell lines treated with or without TA3 using qRT-PCR (Figure 1c). The expression of the HMGCS-1 was increased following TA3 treatment in all three cell lines (1.2-fold in AsPC-1, 1.3-fold in BxPC-3, and 1.6-fold in PANC-1). INSIG-1 mRNA levels were increased 1.6-fold (AsPC-1), 1.9-fold (BxPC-3), and 1.3-fold (PANC-1) in the TA3 treatment compared with the control. After the TA3 treatment, the mRNA level of HMGCR was reduced by 1.2-fold in BxPC-3 cells and 1.8-fold in PANC-1 cells. HMGCR is expressed by SREBP-2 and is involved in the cholesterol synthesis pathway, such as HMGCS-1 [17]. To investigate the interacting INSIG-1 and HMGCS-1 genes, we used GeneMANIA. The results indicated that 20 genes were related to INSIG-1 and HMGCS-1, including SREBF2, SREBF1, SCAP, HMGCR, HMGCS2, and INSIG2 (Appendix A). As mentioned in the Introduction, SREBP bound by INSIG-1 remains an inactive precursor in the ER as the SCAP/SREBP complex. Therefore, we evaluated the expression of SREBPs to determine whether the increase in INSIG-1 caused by TA3 treatment regulates SREBPs. We treated the three cell lines with TA3 (5 µM) for 12 h and then isolated the nuclear and cytosolic fractions. The mature form of SREBP-1 migrates into the nucleus and activates promoters of lipogenic genes [24]. As shown in Figure 1d, the effect of TA3 on the regulation of the expression level of mature SREBP-1 in the nucleus was not significant but was decreased the most in BxPC3 cells among the three cell lines. In the same treatment, protein was extracted from the whole cell. We also monitored the protein levels of the SREBP-1 downstream targets [16], where the regulation of FASN protein levels by TA3 was significantly inhibited only in BxPC-3 cells compared to other cell lines. Another target, the ACC protein level were reduced by the TA3 treatment only in BxPC-3 cells but this effect was not significant (Figure 1e). Taken together, the effect of TA3 on the regulation of SREBP-1 expression, including related genes (HMGCS-1, INSIG-1, FASN, and ACC), was greater in BxPC-3 cells compared to other cell lines.

### 3.2. TA3 Reduced the Mature SREBPs in BxPC-3 Cells

Next, we incubated BxPC-3 cells with TA3 (2.5 and 5 µM) for 12 h and performed Western blotting to confirm whether the expression levels of FASN and ACC, which are two regulators of fatty acid synthase, were regulated by TA3 in a dose-dependent manner (Figure 2a,b). The 5 µM of TA3 caused a significant decrease in the levels of FASN and ACC compared to the untreated control group, although the decrease was not dose-dependent. The expression level of HMGCR, which is an HMG-CoA reductase that converts HMG-CoA to mevalonic acid, was significantly decreased after treatment with TA3 (5 µM) for 12 h. SREBP-2 activates the expression of enzymes, such as HMGCR, that control cholesterol homeostasis [25]. We also observed a significant decrease in the expression of SREBP-2 in the nucleus after treatment with TA3 (2.5 and 5 µM) for 12 h (Appendix A). BxPC-3 cells were incubated with TA3 (2.5 and 5 µM) for 12 h and photographed following immunofluorescent (anti-SREBPs-RFP) and DAPI (nuclear) staining. The immunofluorescence result was consistent with the Western blot result and confirmed that TA3 inhibited the translocation of endogenous SREBP-2 into the nucleus and also reduced its expression in the cytosol (Appendix A) at both concentrations. We also found that SREBP-1 suppressed translocation into the nucleus due to the TA3 treatment and showed weak exposure in the cytoplasm compared to the control (Figure 2c). Similar results were observed using Western blotting, in which the nuclear expression of SREBP-1 was markedly inhibited by TA3 (Figure 2d). To determine whether SREBP-1 regulates apoptosis and cell cycle arrest in BxPC-3 cells, we knocked down the expression of SREBP-1 by transfecting BxPC-3 cells with si-SREBP-1. To evaluate whether si-SREBP-1 induces the knockdown of SREBP-1 in BxPC-3 cells, we performed immunoblotting and immunofluorescence. The results showed that the expression of the SREBP-1 mature form and precursor form in the nucleus was inhibited by si-SREBP-1 compared with scramble si-RNA as a control in BxPC-3 cells (Figure 2e). Although the reduction of SREBP-1 was not significant due to the high deviation obtained from independent triplicate experiments, in the immunofluorescence results, the nuclear localization of SREBP-1 was clearly reduced by the si-SREBP-1 treatment in BxPC-3 cells (Figure 2f). Thus, these results showed that our si-SREBP-1 protocol significantly silenced SREBP-1 in BxPC-3 cells. Silencing of SREBP-1 significantly lowered the expression of lipid-metabolism-related enzymes (FASN and ACC) and increased cleaved PARP (1.4-fold) and p21 (1.9-fold) compared to the control (Figure 2g,h). Although no significant results were obtained due to the high deviation, the expression of p27 was greatly increased 2.9-fold by si-SREBP-1. These results indicated that SREBP-1 played an important regulatory role in lipogenesis, apoptosis, and cell cycle arrest in BxPC-3 cells.

### 3.3. TA3 Increased Cell Cycle Arrest in the G0/G1 Phase and Regulated the Expression of Various Genes Involved in Apoptosis

To determine whether TA3 plays a role in the regulation of cell growth, we treated the cells with various concentrations of TA3 and analyzed the cells for apoptosis and cell cycle arrest using flow cytometry. In the cell cycle assay results, our cell cycle graph represented independent triplicate experiments. Therefore, the results of the cell cycle distribution in Figure 3a showed one of three independent experiments. Then, we graphed the mean of three independent experiments. As shown in Figure 3b, The G0/G1 cell population increased from 61.1% in the control to 65.6% following the TA3 (5 µM) treatment. Additionally, we measured the expression of G0/G1-phase-related genes using qRT-PCR. As shown in Figure 3c, we found that the mRNA levels of genes involved in cell cycle regulation (cyclin E1, D1, and CDK2) were significantly decreased at both TA3 concentrations (2.5 and 5 µM), and the expression of CDK6 was markedly decreased at the high TA3 concentration of 5 µM. However, no changes were observed in the mRNA levels of CDK4 following the TA3 treatment. To determine whether the TA3 treatment-induced G0/G1 cell cycle arrest was dependent on p21 or p27 expression, we analyzed the p21 or p27 protein levels (Figure 3d). The results revealed that TA3-induced G0/G1 cell cycle arrest was dependent on the accumulation of p21 and p27. We examined the number of apoptotic cells after treatment with the indicated concentrations of TA3 (2.5 and 5 µM) for 24 h using Annexin V/FITC double staining. As shown in Figure 3e,f, the percentages of early and late apoptotic cells increased in a dose-dependent manner. We observed the expression of pro-apoptotic genes (cleaved caspase-3, -9, and Bid) and anti-apoptotic genes (Bcl-2) using qRT-PCR (Figure 3g). Consequently, the caspase-9 mRNA level was increased and the Bcl-2 mRNA level was decreased in a TA3-dose-dependent manner. Both genes were significantly regulated in 5 μM. Whereas pro-apoptosis caspase-3 and Bid were not regulated in a TA3-dose-dependent manner, caspase-3 was elevated with 2.5 μM and 5 μM of TA3. Furthermore, both concentrations (2.5 and 5 µM) of TA3 clearly increased the expression of cleaved PARP, which is widely used in apoptotic markers, while cleaved caspase-3 only slightly increased at 5 μM compare to the control (Figure 3g). Taken together, these results demonstrated that TA3 induced cell cycle arrest and apoptosis in BxPC-3 cells.

### 3.4. The Inhibition of SREBP-1 by TA3 Appeared Independent of the Akt/Gsk3β Mechanism

Several studies reported that the phosphorylation of Akt/Gsk3β, mTOR, and AMPK are critical regulators of SREBP-1 [26,27,28]. First, we evaluated the effect on the phosphorylation of Akt, Gsk3β, and mTOR after 12 h of TA3 treatment. Phosphorylation of Akt at Ser473 was slightly increased with 2.5 μM of TA3 but was markedly inhibited by the TA treatment at 5 μM compared with the untreated control. Increased phosphorylation of Gsk3β at Ser9 was increased by the TA3 treatment at both concentrations compared with the untreated control. Gsk3 is known to be an enzyme that phosphorylates and inactivates glycogen synthase [29]. Therefore, our results showed that the TA3 treatment significantly induced Gsk3β inactivation, whereas phosphorylation of mTOR at Ser2448 was not changed by the TA3 treatment compared to the untreated control (Figure 4a,b). In addition, phosphorylation of AMPK was slightly increased following the TA3 (5 µM) treatment. Based on the results, we hypothesized that TA3 inhibits SREBP-1 through the Akt/Gsk3β pathway in BxPC-3 cells. To test this hypothesis, BxPC-3 cells were pretreated with insulin for 30 min to overexpress all the downstream proteins and then treated with or without TA3 for 12 or 24 h (Figure 4c–f). Insulin signaling in various cell models (colorectal, liver, and breast cancer cells) was reported to be a potent transcriptional inducer that regulates SREBP-1 through Akt, mTORC1, and Gsk3β [30,31]. The induction of Akt by insulin was reduced following the TA3 treatment but it was significantly higher than that in cells treated with TA3 alone (Figure 4c). The expression of the Gsk3β protein remained unchanged in the insulin-treated BxPC-3 cells. However, the increased level of the Gsk3β protein after the TA3 treatment was suppressed by insulin pretreatment. In contrast, as shown in Figure 4d, both the matured and precursor forms of SREBP-1 increased following the insulin treatment of the cells. The SREBP-1 protein level was reduced following further treatment with TA3, and the reduction in the level was equal to the SREBP-1 inhibition mediated by the TA3 treatment alone. We confirmed that the FASN levels induced by insulin in BxPC-3 cells were significantly reduced by the TA3 treatment, although the level of the reduction was similar to or higher than the TA3 treatment alone. ACC levels were not induced by insulin in BxPC-3 cells. Nevertheless, the ACC level was significantly decreased by the TA3 treatment with or without insulin (Figure 4e). Interestingly, we found that cleaved PARP and p21 protein levels were slightly decreased in the insulin-treated BxPC-3 cells, whereas the p27 protein level was increased twofold (Figure 4f). We observed that the significantly increased expression levels of p27, p21, and cleaved PARP after treatment with TA3 in the insulin pretreated group did not show a significant difference compared with TA3 treatment alone. Collectively, these results demonstrated that the inhibition of SREBP-1 by TA3 was regulated independently of the Akt/Gsk3β pathway.

### 3.5. The Effect of TA3 on Tumor Growth Reduction Was Greater Than That of GEM in a Xenograft Model

A schematic of the animal experimental protocol is shown in Figure 5a. The volumes of subcutaneous BxPC-3 tumors treated with TA3 or GEM were significantly smaller than those of the PBS-treated vehicle group over 20 days of observation (Figure 5b). On day 20 (final measurement), the tumor volume was significantly reduced in the TA3 treatment group compared to that in the GEM treatment group. At the end of the experiment, the average tumor volumes were markedly lower in the TA3 or GEM treatment groups compared to the vehicle group (Figure 5c,d). In the photograph, the size of the GEM-treated tumor does not seem to show a significant difference from that of the vehicle-untreated tumor, but it was confirmed that the GEM-treated tumor was reduced by measuring the volume because it had a flat shape. To investigate the toxicity of each drug, we measured the body weight (Figure 5e) of the animals. Importantly, compared to the vehicle group, there was a slight body weight loss in the TA3 treatment group, but a greater body weight loss was noted in the GEM treatment group. We also performed an MTT assay by culturing BxPC-3 cells in TA3 and GEM separately or together for 24 h to confirm the cytotoxicity (Appendix A). The cells showed 62% toxicity following the 5 μM TA3 treatment. Furthermore, GEM toxicity (1.25, 2.5, and 5 μM) was found to be dose-dependent (37.4%, 43.8%, 58.7%). Interestingly, the 2.5 μM concentration of TA3 was non-toxic, with cytotoxicity of −2.4%, although when treated with GEM, 49.4% (1.25 μM), 60.6% (2.5 μM), and 72.7% (5 μM) toxicity levels were noted and the toxicity significantly increased by 12%, 16.8%, and 14.0%, respectively, compared to the GEM treatment alone at each concentration. When incubated with a 5 μM concentration of TA3 and GEM (1.25, 2.5, and 5 μM), the toxicity levels in BxPC-3 cells significantly increased by 51.8%, 52.3%, and 41.3%, respectively. Additionally, we examined the expression of apoptosis (PARP and cleaved caspase-3) and cell cycle arrest (p21 and p27)-related proteins in subcutaneous BxPC-3 tumor tissue by IHC staining. The staining results showed that the key apoptotic factors (cleaved PARP and caspase-3) expression were significantly higher in the TA3 treatment group compared to the vehicle group, and p21 and p27 staining were significantly increased in the TA3 treatment group (Figure 5f,g).

### 3.6. Downregulation of Palmitate and Stearate Levels in the Pancreatic Tumor by TA3

Next, we examined whether the reduction in FASN expression had any impact on cellular fatty acid levels. Quantitative analysis of fatty acids in the tumor tissue was performed using LC/MS. First, we used fatty acid standards to set up the analytical separation conditions (Appendix A) and calibration curves (Table 3). As shown in Figure 6a, palmitic and stearic acid levels were significantly decreased in the TA3-treated tumors compared to the vehicle-treated tumors. Therefore, we performed IHC staining to visualize SREBP-1 and FASN expression following the TA3 treatment in subcutaneous BxPC-3 tumor tissue. As shown in Figure 6b, the expression of SREBP-1 and FASN was lower in TA3-treated tumors than in vehicle-treated tumors. This suggested that TA3 exerted its anticancer effects by decreasing the expression of SREBP-1 target genes associated with the biosynthesis of fatty acids in the xenograft mouse model. Moreover, we investigated the effect of TA3 on lipid metabolism in BxPC-3 and 3T3-L1 cells using ORO staining of both cell lines. 3T3-L1 cells are widely used in biological research on adipose tissue, and cell death did not appear, even after 24 h of treatment with TA3 at 5 μM or less, whereas cytotoxicity was observed at concentrations above 10 μM (Appendix A). As shown in Figure 6c,d, the lipid drop content in both cell lines was substantially reduced after treatment with TA3. These results implicated that the inhibition of SREBP-1 in the TA3-treated subcutaneous BxPC-3 tumor reduced lipid metabolism. Expectedly, in vivo data showed potent antitumor effects consistent with in vitro experiment results, and the results of reduced palmitate and stearate level in subcutaneous BxPC-3 tumors supported the SREBP-1 inhibitory effect of TA3 in cell experiments. Thus, we suggest that TA3 has an anti-lipogenic effect through inhibition of SREBP-1 and enzymes involved with lipid metabolism.

## 4. Discussion

Previous studies, including a study by our group, reported the inhibitory effect of TA3 in PC cells [11,13]. However, previous studies lacked profiling data to demonstrate the mechanism underlying the TA3-mediated anticancer effect on PC cells. In this study, we used gene expression profiling to understand the mechanism involved in TA3-mediated inhibitory effects on PC cell growth. Interestingly, our microarray analysis identified the potential regulation of INSIG-1 and HMGCS-1 genes by TA3 in BxPC-3 cells. As both genes are involved in lipid metabolism [32,33] (cholesterol synthesis, fatty acid synthesis, and ketogenesis) our data suggested the potential of TA3 to modulate lipid metabolism. Lipid metabolism is important for tumor growth through the formation of membranes [34]. Several studies have indicated abnormally increased lipid synthesis as one of the main characteristics of PC cells associated with the development of PC malignancy [34,35]. Fritz et al. reported that INSIG-1 is a potential novel tumor suppressor in HCC [36]. INSIG-1 was reported to suppress SREBP-1c expression induced by high glucose [18]. Its binding retains SREBP-1 as an SREBP-SCAP complex on the ER membrane and suppresses its translocation into the nucleus [37]. The abnormalities in the SCAP/SREBP pathway that is involved in lipogenesis are often associated with tumorigenesis and tumor development [18]. Our data showed that the high expression of SREBPs, a family of transcription factors that regulate enzymes of cholesterol in PC [20], was suppressed by TA3 in BxPC-3 cells. Proteolytic activation of nuclear SREBP-1 is under the control of hormone or signal transduction systems, and SREBP-2 nuclear translocation is regulated by the ER cholesterol content, reflecting its strong involvement in cholesterol homeostasis [38]. In the present study, we focused on estimating whether the anticancer activity of TA3 in BxPC-3 cells is related to SREBP-1, which regulates enzymes involved in fatty acid synthesis and understaning the mechanisms. Our data demonstrated that TA3 suppressed mature SREBP-1 and SREBP-2 expression on the nucleus in BxPC-3 cells and reduced the levels of lipid accumulation in 3T3-L1 and BxPC-3 cells. Similar to our results, Sun et al. reported that SREBP-1 was highly expressed in 60 patients with pancreatic ductal cancer [39] and also showed that BxPC-3 cells showed the most sensitivity, with decreased cell viability following silencing of SREBP-1 in BxPC-3, SW1990, and PANC-1 cells [20,40].

Many studies reported that SREBP-1 is associated with cell cycle regulation and apoptosis in cancer [20,34]. Bengoechea-Alonso and Ericsson suggested that siRNA-mediated inactivation of SREBP-1 arrested cells in the G1 phase of the cell cycle, thereby attenuating cell growth [41]. Likewise, our results of the cell cycle analysis showed that TA3 treatment induced G0/G1 phase arrest and inhibited various cyclin-dependent protein kinases (Cdks) associated with G0/G1 phase checkpoints. The levels of p21 and p27 were increased after treatment with TA3 in BxPC-3 cells. Furthermore, our results showed that TA3 treatment led to increased apoptotic cell death, as evidenced by increased levels of cleaved PARP and caspase-3. The level of cleaved PARP was increased following the silencing of SREBP-1 in BxPC-3 cells, and this result was the same as that observed in TA3-mediated apoptosis, which suggested that the cell growth inhibitory effect of TA3 was dependent on the expression of SREBP-1. These findings are in close agreement with the results of Li et al., who showed that knockdown of SREBP-1 expression in endometrial cancer cells suppressed cell growth, reduced colonigenic capacity, and slowed tumor growth in vivo [42]. Our data provided significant insights into the critical role and mechanisms of TA3 via blocking the maturation and translocation of SREBPs, which represents a promising approach for perturbing the cancer metabolic program and attenuating cancer growth.

Several studies reported that PI3K/Akt signaling controls metabolic flux from glucose and glutamine to lipid synthesis [27]. SREBP-1 is stabilized and activated by the PI3K/Akt signaling pathway in cancer [43]. Fwb-7 mediates SREBP-1 N-terminal degradation and is activated by active GSK3β, which is inhibited by PI3K/Akt signaling [28]. An alternate PI3K/Akt signaling pathway involves mTORC1, which regulates phosphatase lipin 1 to control SREBP-1 nuclear localization and transcriptional activity [19]. Our results showed that SREBP-1 might be reduced through the activation of GSK3β by TA3. Other studies showed that AMPK interacts with SREBP-1 and -2 or directly phosphorylates them [26]. AMPK stimulates Ser372 phosphorylation, inhibits SREBP-1 cleavage and nuclear translocation, and inhibits SREBP-1 target gene expression in hepatocytes exposed to high glucose [44]. Our results show that TA3 slightly stimulates AMPK, which may help to reduce fat production and lipid accumulation. Therefore, we attempted to demonstrate that the Akt/GSK3β signaling pathway is a key inhibitor of SREBP-1 by TA3. Reductions in SREBP-1, FASN, and ACC expression were shown in the TA3 treatment after insulin pretreatment as much as with TA3 treatment alone. The apoptosis-related factors (cleaved PARP and caspase-3) and cell cycle arrest-related factors (p21 and p27) showed the same trend, whereas the suppression of Akt phosphorylation and induction of GSK3β inactivation due to increased Ser9 phosphorylation after the TA3 treatment was blocked by insulin. These results clearly suggested that TA3 reduced cell growth through SREBP-1 inhibition and was not dependent on the inhibition of Akt/GSK3β signaling.

GEM is a first-line chemotherapeutic agent for PC. However, many patients do not respond to the drug because of both intrinsic and acquired resistance [45]. In recent years, interest in natural extracts as anticancer agents has been of considerable interest as they offer the potential to discover many active compounds and significantly reduce chemotherapy-related toxicities by replacing them with naturally derived ingredients [46]. Interestingly, in the animal experiments, TA3 seemed to be more potent than GEM as an inhibitor of the xenograft growth of BxPC-3 cells. Therefore, given that the reduction in body weight, as an indicator of toxicity of agents, was more in the GEM group than in the TA3 group, TA3 seemed to be safer than GEM. Furthermore, consistent with the in vitro results, we demonstrated that fatty-acid-producing enzymes and SREBP-1 transcription factors were inhibited in the TA3 group. Palmitic acid is the most abundant saturated fatty acid in nature and is the precursor for a variety of long-chain fatty acids, such as stearic acid, palmitoleic acid, and oleic acid. The fatty acid synthesis starts with the formation of palmitic acid from acetyl-CoA and malonyl-CoA [47,48]. We quantitatively confirmed that palmitic acids and stearic acid in the PC tissues decreased in the treatment group compared with the vehicle group, which clearly supported the in vitro results. Similarly, Kwan et al. reported lower levels of palmitic acid (the end product of FASN) and stearic acid, suggesting an inhibitory oridonin effect on FASN activity and cell growth in colorectal cancer cells [49]. Another study reported that intermediate metabolites containing C16:0, C18:0, and C18:1n9 were consistently decreased by SREBP-1 knockdown [50]. Enhanced fatty acid biosynthesis is crucial to meet the demand for lipids for membrane synthesis during cell proliferation [51]. These data suggest that TA3 decreases the expression of cancer-associated lipogenic genes and further reduces intracellular fatty acids to inhibit PC growth in vitro and in vivo.

In summary, our findings suggested that TA3 affected PC cell growth by inhibiting SREBP-1 expression. This study provides the first evidence demonstrating that TA3 inhibits the level of palmitate and stearate, and it induces apoptosis in BxPC-3 cells. Mechanistically, although the suppression of Akt phosphorylation and induction of GSK3β-Ser9 inactivation by TA3 was blocked by pretreatment with insulin, the inhibitory effect of mSREBP-1, FASN, and ACC expression by TA3 was maintained, even with pretreatment with insulin as much as when treated with TA3 alone. Thus, our results established that the TA3 inhibitory effect on SREBP-1 was independent of the regulation of Akt and GSK3β phosphorylation. Taken together, we propose that TA3 might be a direct inhibitor of SREBP-1 that is capable of inducing the inhibition of fatty acid synthesis and tumor growth. Our findings provide an understanding of the pharmacological mechanism of TA3 and indicate its immense potential as a target for deficiency in lipid metabolism for PC therapy.

## Figures and Tables

**Figure 1 pharmaceutics-14-00945-f001:**
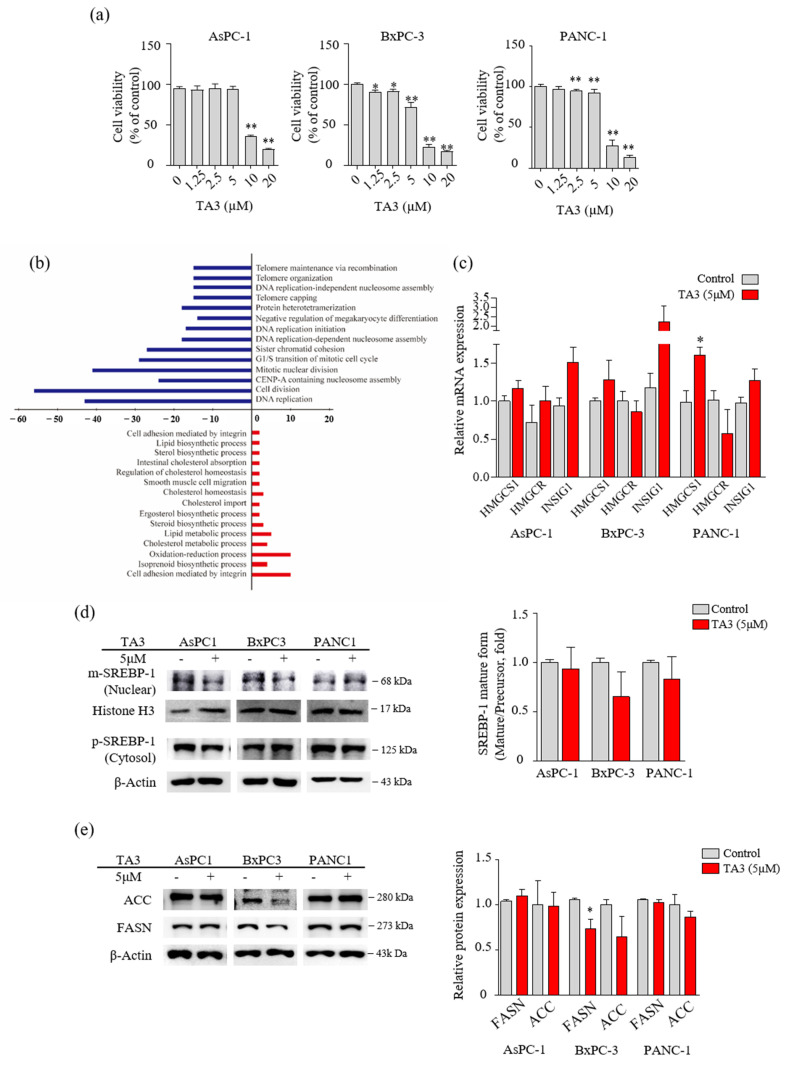
TA3 affected the regulation of enzymes involved in lipid metabolism in BxPC3 cells. (**a**) The cells were treated with TA3 at various concentrations for 24 h. The cell viability was measured using an MTT assay. (**b**) Differentially expressed oncologic genes presented a distribution of biologic functions when using 5 μg/mL TA3 on BxPC-3 cells. Up- and downregulated genes were grouped into 15 and 14 biologic functional categories based on the microarray annotation system. The *x*-axis represents the number of gene transcripts for the 1.5-fold change regulation in each category. Negative values (blue bar) represent downregulated genes and positive values (red bar) represent upregulated genes after the TA3 treatment. (**c**) RNA expression for the indicated genes was measured using qRT-PCR in the PC cell lines. Same as the microarray, the three cell lines were treated with 5 μM TA3 for 24 h. (**d**,**e**) To evaluate the expression of proteins, the three cell lines were incubated with TA3 (5 μM) for 12 h. (**d**) The extracts separated into nuclear and cytoplasm were prepared and the lysates were used for immunoblotting to measure the expression of precursor and mature SREBP-1 (pSREBP-1 and mSREBP-1). (**e**) The effects of TA3 on the FASN and ACC expression levels were identified in the total proteins of three cell lines using immunoblotting. All assay results are indicated in the bar graph as fold changes compared with that of the control as the mean ± SD of three independent experiments (* *p* < 0.05, ** *p* < 0.01, vs. control).

**Figure 2 pharmaceutics-14-00945-f002:**
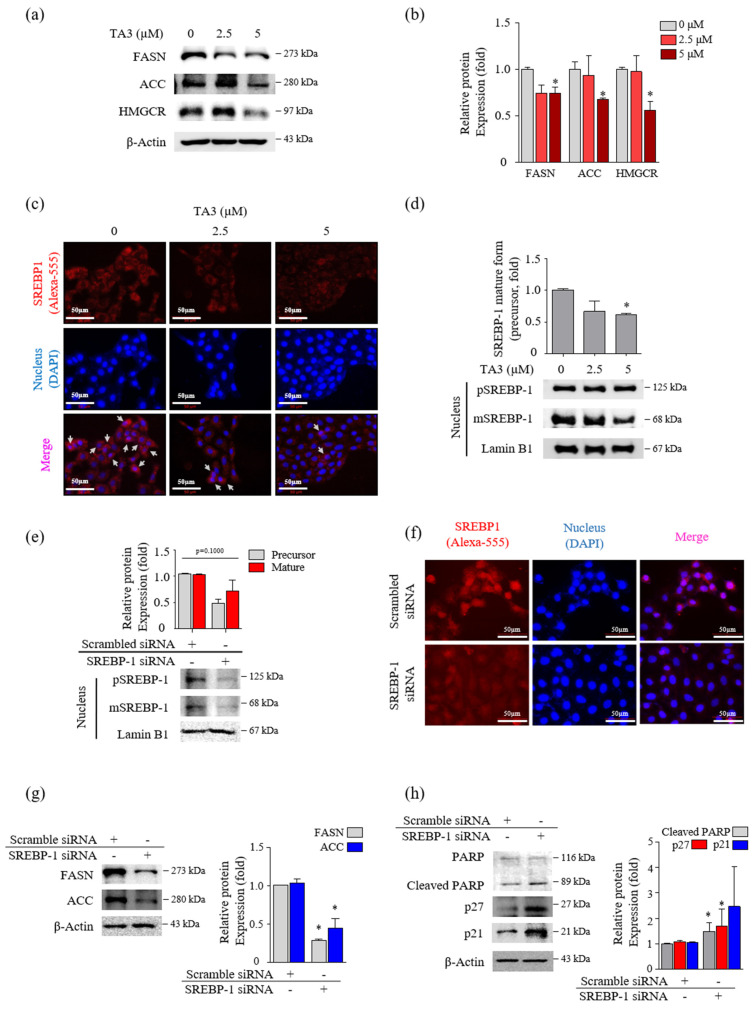
TA3 reduced the nuclear localization of mature SREBP-1, which inhibited the lipogenic enzymes and cell growth in BxPC-3 cells. (**a**–**d**) The effect of TA3 treatment for 12 h on SREBP-1 transcriptional activity and lipogenic enzymes expression in BxPC-3 cells. (**a**) The SREBP-1 target genes encoding ACC, FASN, and HMGCR were examined via immunoblotting analysis. (**b**) The intensities are presented in the bar graph as fold changes compared with that of the control (* *p* < 0.05, vs. control). (**c**) An immunofluorescence assay was performed to image analyze the location of SREBP-1 expression in BxPC-3 cells. Arrows denote nuclear translocation. (**d**) The expressions of pSREBP-1 and mSREBP-1 levels were determined via immunoblotting in nuclear extracts. (**e**–**h**) The SREBP-1 silencing efficiency of 24 h transfection of SREBP-1 siRNA in BxPC-3 cells. After transfection, the protein was isolated and analyzed via (**e**) immunoblotting and (**f**) immunofluorescence assays to evaluate the expression of SREBP-1 in the nucleus. (**g**,**h**) The expressions of proteins on the silencing of SREBP-1 were analyzed by immunoblotting in BxPC-3 cells. The immunoblotting intensity represents the mean ± SD of triplicate samples (* *p* < 0.05, vs. control). The magnification of the image is 400×. Scale bar, 50 μm.

**Figure 3 pharmaceutics-14-00945-f003:**
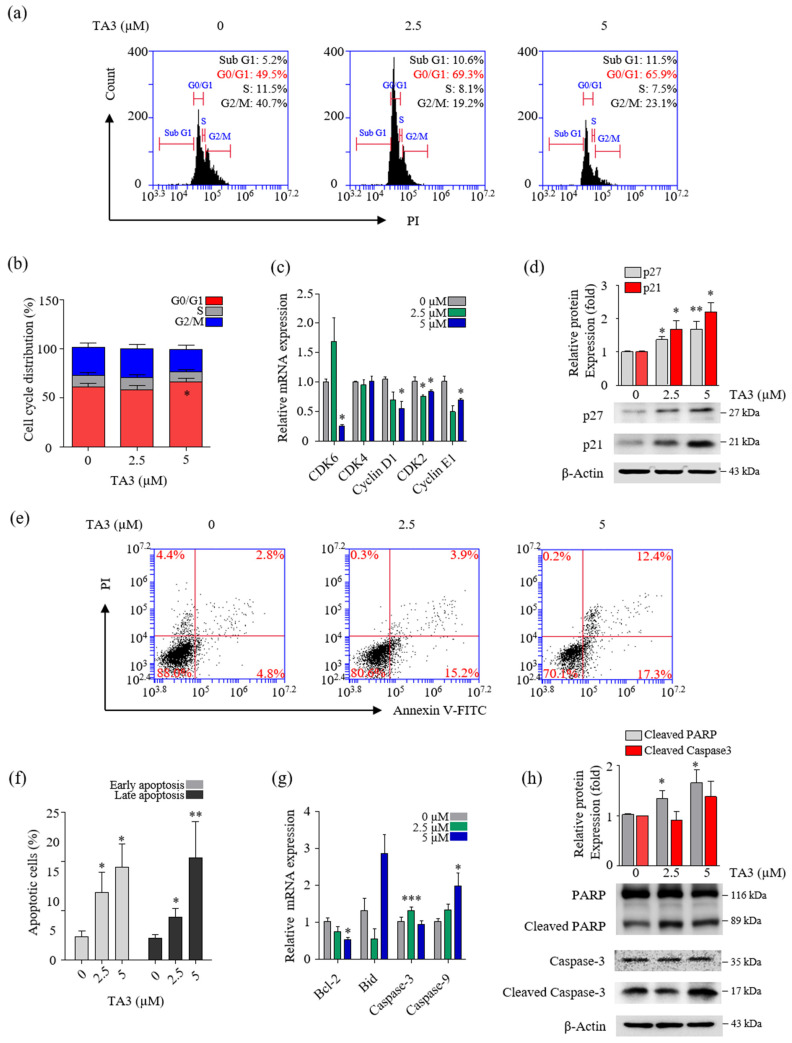
TA3 inhibited proliferation in BxPC-3 cells. The BxPC-3 cells were incubated with TA3 for 24 h in all assays. (**a**) Flow cytometry was used to analyze PI-stained cells and (**b**) the results were presented in the bar chart as the mean ± SD of the independent three experiments (* *p* < 0.05 vs. control). (**c**) The G1/G0-phase-related genes’ mRNA levels were analyzed using qRT-PCR. (**d**) The effects of TA3 on the p27 and p21 expression levels related to cell cycle arrest were detected using immunoblotting. (**e**) The apoptotic status of cells was depicted using dot plots. (**f**) Flow cytometry data are expressed in the bar graph as the mean ± SD of the independent three experiments. (**g**) The expressions of various gene levels associated with apoptosis were analyzed using qRT-PCR. (**h**) The effects of TA3 on the PARP and caspase-3 expression levels were identified using immunoblotting. All results are presented as fold changes compared with that of the control as the mean ± SD of three independent experiments (* *p* < 0.05, ** *p* < 0.01, *** *p* < 0.001 vs. control).

**Figure 4 pharmaceutics-14-00945-f004:**
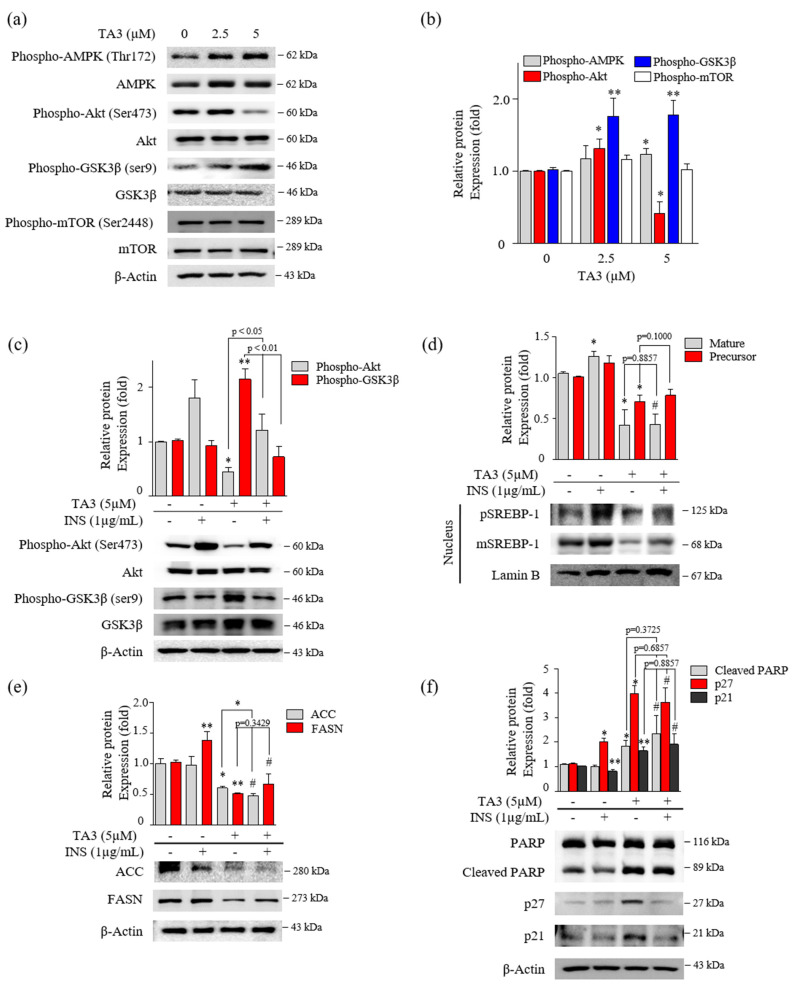
Inhibition of SREBP-1 expression by TA3 was independent of Akt and Gsk3β regulation. (**a**) The regulation of AMPK, Akt, Gsk3β, and mTOR expression on BxPC-3 cells treated with TA3 for 12 h. (**b**) Immunoblotting results are presented in the bar graph as fold changes compared with that of the control as the mean ± SD of three independent experiments. (**c**–**f**) Activation of proteins was induced by insulin and compared with/without TA3 treatment. After 12 h, extracted proteins were analyzed using immunoblotting, and the expressions of proteins were observed using (**c**) Akt and GSK3β antibodies. (**d**) The expressions of SREBP-1 and (**e**) their downstream target proteins, including FASN and ACC, were examined via immunoblotting analysis. (**f**) The apoptosis and proliferation-related protein expression levels were identified using immunoblotting. All data are expressed as the mean ± SD of three independent experiments (* *p* < 0.05, ** *p* < 0.01, vs. control, and ^#^
*p* < 0.05 vs. INS (1 µg/mL)).

**Figure 5 pharmaceutics-14-00945-f005:**
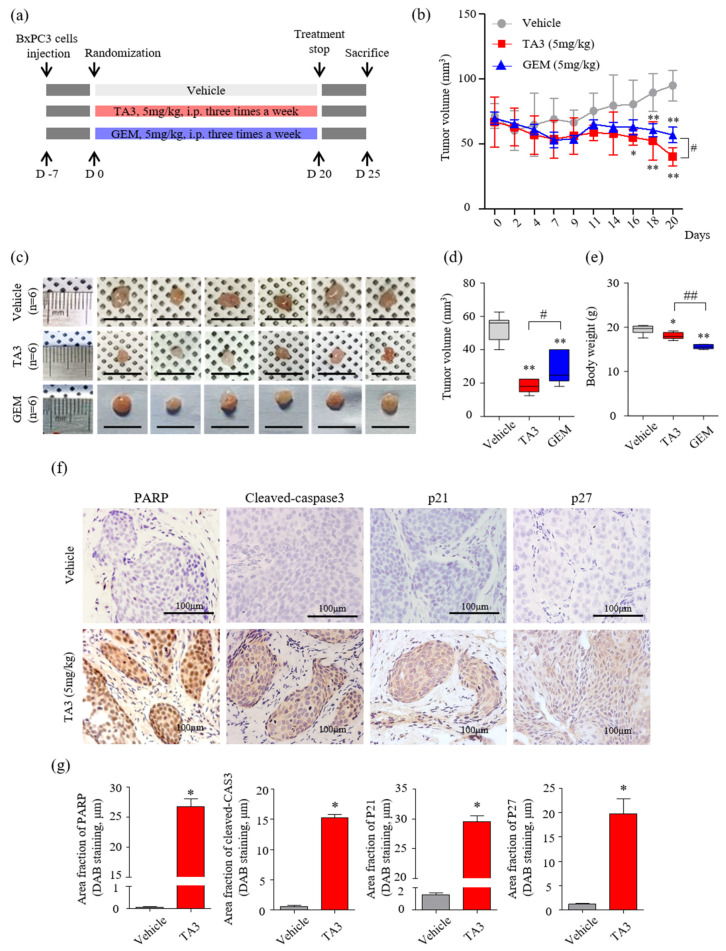
Effects of TA3 on pancreatic cancer growth in a xenograft mouse model. (**a**) Experimental methods of schematic representation as described in the Materials and Methods section. Briefly, BxPC-3 cells (1 × 10^7^ cells/mouse) were mixed with Matrigel at a ratio of 1:1 (*v/v*) before subcutaneous injection and randomly assigned to six mice of each group. Then, 100 μL of PBS and each drug (5 mg/kg of TA3 and GEM) were intraperitoneally injected into the mice of each group three times a week for 20 days with a measure of tumor volume and euthanized five days after the end of treatment. (**b**) Tumor volume was monitored for three weeks after treatment with TA3 (5 mg/kg). (**c**) Tumors harvested from each treatment group were photographed. (**d**) The volume of the dissected primary tumor was calculated using the formula V = 4/3 πr^3^. (**e**) Estimated cytotoxicity using the measured body weight. (**f**) Paraffin-embedded tumor sections were stained brown through immunoreactivity with PARP, cleaved caspase-3, p21, and p27 antibodies using immunohistochemical analysis. (**g**) Comparison between the immunostaining (DAB) area fraction of PARP, cleaved caspase-3, p21, and p27 in paraffin-embedded tumor sections. All box plot presented the mean ± SD (* *p* < 0.05, ** *p* < 0.01, vs. vehicle; ^#^
*p* < 0.05, ^##^
*p* < 0.01 comparison of TA3 and GEM; *n* = 6/group).

**Figure 6 pharmaceutics-14-00945-f006:**
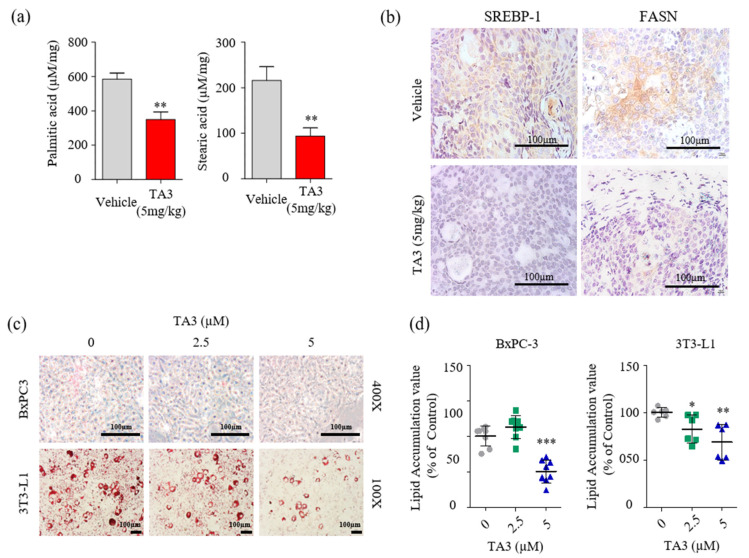
TA3 inhibited SREBP-1 and FASN expressions, which were involved in the lipogenesis pathway, and lowered fatty acid levels in pancreatic tumor tissues. (**a**) Comparison of concentrations of palmitic acid and stearic acid in the vehicle and TA3 groups’ tumor tissues using LC/MS. The data are shown in the bar graph as the mean ± SD (* *p* < 0.05, ** *p* < 0.01, vs. vehicle; *n* = 6/group). (**b**) Paraffin-embedded tumor sections were incubated with SREBP-1 and FASN antibodies and stained brown using a DAB solution. (**c**) BxPC-3 and 3T3-L1 cells were pretreated with TA3 (2.5 and 5 µM) for 48 h. Intracellular lipids accumulation was stained red using filtered Oil Red O solution. After staining, both cells were photographed and quantified using an automated microplate ELISA reader. Scale bar, 100 µm. (**d**) The value presented as a percentage relative to that of the control as the mean ± SD of three independent experiments (* *p* < 0.05, ** *p* < 0.01, *** *p* < 0.001, vs. control).

**Table 1 pharmaceutics-14-00945-t001:** List of primer sequences used for qRT-PCR.

Primers	Forward (5′→3′)	Reverse (5′→3′)
*CDK6*	TCTTCATTCACACCGAGTAGTGC	TGAGGTTAGAGCCATCTGGAAA
*CDK4*	ATGGCTACCTCTCGATATGAGC	CATTGGGGACTCTCACACTCT
*CDK2*	AAAGCCAGAAACAAGTTGACG	GAGATCTCTCGGATGGCAGT
*Cyclin D1*	TTCGATGATTGGAATAGC	TGTGAGCTGCTCATTGAG
*Cyclin E1*	GAAATGGCCAAAATCGACAG	TGTCAGGTGTGGGGATCA
*Bcl-2*	GATGGCAAATGACCAGCAGA	GCAGGATAGCAGCACAGGAT
*Bid*	ATGGACTGTGAGGTCAACAACGG	CACGTAGGTGCGTAGGTTCTGGTTA
*Caspase-3*	AGCAAACCTCAGGGAAACATT	GTCTCAATGCCACAGTCCAGT
*Caspase-9*	GGTTCTGGAGGATTTGGTGA	GACAGCCGTGAGAGAGAATGA
*HMGCS1*	CTCCCTGACGTGGAATGTCT	GAACTGTCTGCCCAGGTGAT
*HMGCR*	CTTGCCGAGCCTAATGAAAG	TGACCCCCTGAGAAAGCTAA
*INSIG1*	CAACACCTGGCATCATCG	CTCGGGGAAGAGAGTGACAT

**Table 2 pharmaceutics-14-00945-t002:** The 22 genes most differentially expressed in TA3 treatment compared to untreated BxPC-3 cells were selected based on microarray results.

Probe Set ID	Gene Accession	Gene Description	Gene Symbol	Log Ratio (TA vs. Ctr)	Fold Change (TA vs. Ctr)
**Lipid metabolic process**					
16995890	NM_001098272	3-Hydroxy-3-methylglutaryl-CoA synthase 1 (soluble)	*HMGCS1*	2.175	4.51
17053892	NM_005542	Insulin-induced gene 1	*INSIG1*	2.043	4.12
16708249	NM_005063	Stearoyl-CoA desaturase (delta-9-desaturase)	*SCD*	1.917	3.78
16972155	ENST00000261507	Methylsterol monooxygenase 1	*MSMO1*	1.729	3.32
16741501	ENST00000355527	7-Dehydrocholesterol reductase	*DHCR7*	1.616	3.07
**Cell cycle**					
16677201	NM_016448	Denticleless E3 ubiquitin protein ligase homolog (Drosophila)	*DTL*	−2.009	4.03
16702571	NM_182751	Minichromosome maintenance complex component 10	*MCM10*	−1.885	3.69
16685165	NM_022111	Claspin	*CLSPN*	−1.727	3.31
16850477	NM_001071	Thymidylate synthetase	*TYMS*	−1.727	3.31
17067332	ENST00000305188	Establishment of cohesion 1 homolog 2 (S. cerevisiae)	*ESCO2*	−1.540	2.91
16877019	ENST00000360566	Ribonucleotide reductase M2	*RRM2*	−1.536	2.90
16703478	NM_001172303	Microtubule-associated serine/threonine kinase-like	*MASTL*	−1.527	2.88
16965346	NM_022346	Non-SMC condensin I complex, subunit G	*NCAPG*	−1.519	2.87
16844312	NM_001067	Topoisomerase (DNA) II alpha 170 kDa	*TOP2A*	−1.519	2.87
17079293	NM_057749	cyclin E2	*CCNE2*	−1.514	2.86
**Cell proliferation**					
17053892	NM_005542	Insulin-induced gene 1	*INSIG1*	2.043	4.12
**Apoptosis**					
16685165	NM_022111	Claspin	*CLSPN*	−1.727	3.31
16844312	NM_001067	Topoisomerase (DNA) II alpha 170 kDa	*TOP2A*	−1.519	2.87
**Cell division**					
16703478	NM_001172303	Microtubule-associated serine/threonine kinase-like	*MASTL*	−1.527	2.88
16965346	NM_022346	Non-SMC condensin I complex, subunit G	*NCAPG*	−1.519	2.87
16844312	NM_001067	Topoisomerase (DNA) II alpha 170 kDa	*TOP2A*	−1.519	2.87
17079293	NM_057749	cyclin E2	*CCNE2*	−1.514	2.86

**Table 3 pharmaceutics-14-00945-t003:** Calibration curve of palmitic acid and stearic acid standards using LC/MS.

Compounds	Formula	RT (min)	Equation
Palmitic acid (C16:0)	C16H32O2	15.18	y = 0.001675x + 1.983431
Stearic acid (C18:0)	C18H36O2	16.24	y = 0.002015x + 2.559123

## Data Availability

All data that support the findings of this study are available from the corresponding author, upon reasonable request.

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
