# Peer review of "Timosaponin A3 Inhibits Palmitate and Stearate through Suppression of SREBP-1 in Pancreatic Cancer"

_pharmaceutics, 2022, doi:10.3390/pharmaceutics14050945_

Round 1
Reviewer 1 Report
Kim et al in the manuscript “Inhibition of SREB-1 transcriptional activity by timosaponin A3 induces fatty acid biosynthesis deficiency and growth inhibition in pancreatic cancer”, show the effect of TA3 on cell growth of pancreatic cancer cells. In particular they focus on the effect of TA3 on BxPC3 cells, since in their previous publications, the effect of TA3 on other pancreatic cancer cells was already shown. Indeed, for instance they have shown that TA3 treatment in AsPC-1 cells could induce apoptosis, suppress cell proliferation by inhibiting the STAT3 and ERK1/2 pathways and induce accumulation in G2/M of the cell cycle. Here, they expand the previous observation by analyzing BxPC3 cells, one of the most used pancreatic cancer cell model expressing a K-ras wild type. I would like to remind to the Authors that almost 95% of pancreatic tumors show a mutant Kras.
Anyhow, using TA3 they performed several experiments in order to show that TA3 induces cell growth arrest, apoptosis, especially because is able, in this specific cell line, to regulate negatively lipogenesis. However, reading the manuscript several concerns arise
- Line 55: The sentence is not clear, maybe it is truncated.
- Line 59: PI3K, Akt and ERK cannot considered transcriptional factors
- 1C: HMGCS1 and INSIG1 are not significantly induced in AsPC1, BxPC3 and only HMGCS1 mRNA is significantly induced in PANC1. Why the Authors claim that these mRNAs are increased following TA treatment?
- The mRNA data are completely unrelated to microarray analysis since in latter both mRNAs are induced more than 4-folds (most regulated ones) while in qPCR analysis they are almost unchanged (see the figure 1C).
- In Figure 1d, the mature form of SREBP is not significantly regulated, indeed also in BxPC3 no statistical significance has been shown by the authors.
- Fig 1e, the authors show a really slight drop in expression of FASN and phosphoACC
- Figure 2g: normally for phosphorylated proteins is necessary to show the ratio between the Ph/total-protein. Here it is not clear how they calculate the phosphor-ACC level
- Figure 2h: p21 expression is not significantly regulated, why they claim an increased expression in the main text? Anyhow, also PARP and p27 changes are very little.
- Fig 3a: why no increase in G1 in TA3 (2,5uM) taking in account that also in this condition they observe p27 and p21 increase. How they explain this contradictory result?
- Fig 3g: usually to show apoptosis it is not useful to analyze gene expression, indeed in this figure Bid is down-regulated and then upregulated, caspase 3 has an opposite trend. How the Authors interpret these variable data?
- In Figure 3h I do not see caspase 3 activation, but the author talk about early and late apoptosis, how these cells die if caspase 3 it is not cleaved or more expressed?
- The take home message of the paragraph 3.4 is quite complicate to appreciate. Maybe was more easy to show that a PI3K or an AKT inhibitor di not interfere with TA3 effect on srebp1 without use Insulin, that increase the level of complexity without lead to new information.
- The authors may explain why p27 is upregulated by insulin since previous data indicate that insulin stimulates proliferation in BxC3 cells (see i.e. Metformin disrupts crosstalk between G protein-coupled receptor and insulin receptor signaling systems and inhibits pancreatic cancer growth.)?
- In xenograft experiments the control tumor grow very slowly, probably because BxPC3 are not the best model to see pancreatic tumor, and therefore it is quite difficult appreciate the difference between control and treated mice. Could be useful to treat longer time the mice before the sacrifice to see if a great difference between the control and the treated mice can be observed.
- In addition, in the first 7 days there is an evident reduction of the tumor volume upon TA3 and GEM treatment, however, at later time points (9-11 days) the same tumors start to grow again and then (14-20 days) they stop again. Longer time will help to see if a real difference does exist between the untreated and the treated one and above all between Gem and TA3 treated mice.
- Fig5f: in vitro data did not show caspase 3 activation while in xenograft is observed, how they explain?
Author Response
Feb 11th, 2022
Manuscript ID: pharmaceutics-1564084
Title: Inhibition of SREBP-1 Transcriptional Activity by Timosaponin A3 Induces Fatty Acid Biosynthesis Deficiency and Growth Inhibition in Pancreatic Cancer
Dear Editors,
We would like to thank you and the reviewers of the Pharmaceutics for taking the time to review our manuscript. We have made corrections and clarifications in the manuscript after going over the reviewer’s comments.
In this revised manuscript, we carefully responded to each reviewer’s comments, point by point, as in the Author’s response. We hope the revised manuscript will better meet the requirements of your journal for publication. We thank the editors and the reviewers of the Pharmarceutics once again for the constructive review of our paper.
Sincerely yours,
Hyeung-Jin Jang
Responses to the reviwers’s comments:
Reviewer 1.
- Line 55: the sentence is not clear, maybe it is truncated.
Our response: Thanks for your comments. As pointed out by the reviewer, we had rewritten in the introduction part. (Line 55, page 2)
- Line 59: PI3K, Akt, and ERK cannot considered transcriptional factors
Our response: Thank you for catching the detailed points. As your comment, we revised the sentence. (Line 57-59, page 2)
- 1C: HMGCS1 and INSIG1 are not significantly induced in AsPC1, BxPC3 and only HMGCS1 mRNA is significantly induced in PANC1. Why the authors claim that these mRNAs are increased following TA3 treatment?
Our response: As pointed out by the reviewer, we revised the sentence. To add an explanation, HMGCS-1 and INSIG-1 mRNA levels were increased by TA3 treatment compare control even it is not significantly. We repeated the independent experiment, and although p value less than 0.05 was not satisfied due to deviation, HMGCS1 in TA3 treatment was increased 1.2-fold (AsPC1), 1.3-fold (BxPC3), and 1.6-fold (PANC1) compared with control. In addition, INSIG-1 was increased 1.6-fold (AsPC1), 1.9-fold (BxPC3), and 1.3-fold (PANC1) in the TA3 treatment group. (Line 275-284, page 6-7)
- The mRNA data are completely unrelated to microarray analysis since in latter both mRNAs are induced more than 4 folds (most regulated ones) while in qRCR analysis they are almost unchanged (see the figure 1C).
Our response: In microarray during the cRNA formation we amplify the available cDNA, so in microarray we cannot exactly measure the quantity of starting mRNA, only it can give you clue for up and down expression. While in qPCR the exact estimation takes place, so there is no need of any correlation coefficient to make both microarray and qPCR equal. Thus, we quantified mRNA level and showed that HMGCS-1 (1.3-fold) and INSIG-1 (1.9-fold) were increased by TA3 using qRCR. And this data purpose, we confirmed a lot of genes expression by microarray and presented effect genes in TA3. As you can see gene network, those two genes are relative with SREBPs.
- In Figure 1d, the mature form of SREBP is not significantly regulated, indeed also in BxPC3 no statistical significance has been shown by the authors.
Our response: We agree with the reviewer about that, However, the goal of this results is to find the connection between SREBP-1 and TA3 in pancreatic cancer cells and which cell is most regulated by TA3. We suggest that SREBP-1 expression was decreased 0.9-fold (AsPC-1), 0.6-fold (BxPC3), and 0.8-fold (PANC-1) by TA3 in Fig 1d. (Line 286-288, page 7)
- Fig 1e, the authors show a really slight drop in expression of FASN and phosphoACC.
Our response: There is some reason in what the reviewer pointed out. We have corrected expression of FASN and phosphoACC in Fig 1e.
- Figure 2g: normally for phosphorylated proteins is necessary to show the ratio between the Ph/total-protein. Here it is not clear how they calculate the phosphor-ACC level
Our response: We have added the sentences for the how they calculate the phosphor-ACC and FASN in immunoblotting method. (Line 159-161, page 4)
- Figure 2h: p21 expression is not significantly regulated, why they claim an increased expression in the main text? Anyhow, also PARP and p27 changes are very little.
Our response: Thanks for your comments. We have revised the sentences in the revised manuscript as reviewer pointed out as follows: Silencing of SREBP-1 significantly inhibited lipid metabolism-related enzymes, and in-creased the expression of cleaved PARP (1.4-fold) and p21 (1.9-fold) compared to control (Fig. 2g and h). Although no significant results were obtained due to high deviation, the expression of p27 was greatly increased 2.9-fold by si-SREBP-1. (Line 330-333, page 9)
- Fig3a: why no increase in G1 in TA3 (2.5uM) taking in account that also in this condition they observe p27 and p21 increase. How they explain this contradictory result?
Our response: As shown in cell cycle assay results, sub G1 was increased by TA3 treatment, which mean that it might be influence the G1 phase by inducing apoptosis. Also, in the referenced paper, the cell cycle results at the same simvastatin concentration, which p21 (1.89-fold) and p27 (1.26-fold) were increased in the western blot experiment did not show that G0/G1 phase was increased compared to control in HepG2 cells (Wang et al. Cell Death Dis 2017 23;8(2):e2626)
- Fig 3g: usually to show apoptosis it is not useful to analyze gene expression, indeed in this figure Bid is down-regulated and then upregulated, caspase 3 has an opposite trend. How the authors interpret these variable data?
Our response: First of all, I am sorry for making you confused. The scope of this study was to determine whether pro-apoptosis and anti-apoptosis are regulates by TA3 dose-dependent. Therefore, we are pleasure to the reviewer’s comments, we looked at the data and found that the cleaved caspase-3 and caspase-3 were swapped in the process of organizing the figures. Thus, we have corrected the expressions in the revised manuscript as reviewer pointed out as follows: We observed the expression of pro-apoptotic genes (Cleaved Caspase-3, -9, and Bid) and anti-apoptotic genes (Bcl-2) by qRT-PCR (Fig. 3g). Consequently, Caspase-9 mRNA level was increased and Bcl-2 mRNA level was decreased in TA3 dose-dependent manner. Both genes were significantly regulated in 5 μM. Whereas, pro-apoptosis caspase-3 and Bid were not regulated in a dose-dependent manner of TA3, and Caspase 3 was elevated at 2.5 μM, and 5 μM of TA3, respectively. Furthermore, both concentrations (2.5 and 5 µM) of TA3 clearly increased the expression of cleaved PARP, widely used in apoptotic markers, while cleaved caspase-3 only slightly increased at 5 μM compare to control (Fig. 3g). (Line 365-373, page 11)
- In Figure 3h I do not see caspase 3 activation, but the author talks about early and late apoptosis, how these cells die if caspase it is not cleaved or more expressed?
Our response: As reviewer pointed out, we looked at the data and we found that the cleaved caspase-3 and caspase-3 were swapped in the process of organizing the figures. Thus, we have corrected the expressions in the revised manuscript. When we submitted the original images for blot document was correct, so we can be verified with it. Unfortunately, although the pharmacological mechanism for caspase-3 expression cannot be completely explained, the results of sub-G1 increase in cell cycle assay, annexin-V assay, and cleaved PARP increase sufficiently demonstrated that TA3 induces apoptosis in BxPC-3.
- The take home message of the paragraph 3.4 is quite complicate to appreciate. Maybe was more easy to show that a PI3K or an AKT inhibitor did not interfere with TA3 effect on SREBP1 without use insulin, that increase the level of complexity without lead to new information.
Our response: We understand with the reviewer pointed out, there is a way to directly silence PI3K or AKT, however we used insulin to broadly identify the pathway TA3 has to inhibit SREBP1. In addition, the role of insulin on SREBP1 progression has been reported in several literatures. Significantly in our results, even under insulin induction, the SREBP1 inhibitory effect of TA3 was the same as the SREBP1 inhibitory effect of a single TA3 treatment. (Line 398-400, page 13)
- The authors may explain why p27 is upregulated by insulin since previous data indicate that insulin stimulates proliferation in BxPC3 cells (see i.e. Metformin disrupts crosstalk between G protein-coupled receptor and insulin receptor signaling systems and inhibits pancreatic cancer growth.)?
Our response: Thanks for your comments. Unfortunately, we could not explain whay p27 is upregulated by insulin. Although, there not many studies on the cellular processes between p27 and insulin treatment, Philippe et al. have suggested that p27 expression was increased as insulin treatment time increased in serum-starved CHO/sst2 cells (JBC 1999; 274; 21:P15186-15193).
- In xenograft experiments the control tumor grow very slowly, probably because BxPC3 are not the best model to see pancreatic tumor, and therefore it is quite difficult appreciate the difference between control and treated mice. Could be useful to treat longer time the mice before the sacrifice to see if a great difference between the control and the treated mice can be observed.
Our response: We agree with the reviewer about that. Unfortunately, we were unable to treat the mice in the control and treatment groups for a longer period of time to obtain sufficient tumor volume for the experiment.
- In addition, in the first 7 days there is an evident reduction of the tumor volume upon TA3 and GEM treatment, however, at later time points (9-11 days) the same tumors start to grow again and then (14-20 days) the stop again. Longer time will help to see if a real difference does exist between the untreated and the treated one an above all between GEM and TA3 treated mice.
Our response: We fully understand with the reviewer pointed out. We wanted to keep the experimental period longer for presented to differences between the control and treated groups could be more clearly shown. However as can be seen in the Fig 5e and d, the volume of the tumors of TA3 and GEM-treated mice was significantly reduced in end points, if we keep the experimental period longer, fatty acid analysis and IHC staining would not have been possible.
- Fig5f: in vitro data did not show caspase 3 activation while in xenograft is observed how they explain?
Our response: Sorry for any inconvenience. We made a mistake and corrected it. As we mentioned question number 10, we looked at the data and found that the cleaved caspase-3 and caspase-3 were swapped in the process of organizing the figures. Thus, we have corrected the expressions in the revised manuscript as reviewer pointed out as follows: “cleaved caspase-3 only slightly increased at 5 μM compare to control (Fig. 3g).” therefore, increased expression of cleaved caspase-3 at TA3 group observed in xenograft model is convincing.

Reviewer 2 Report
Dear Authors,
Thank you for interesting study. I hope this comments find you well. This study shows interesting data concerning pancreatic cancer and potential therapy with TA3. Especially, reduction of tumors volumes in vivo with TA3 treatment. Despite that the reduction of cancer in vivo is very close to GEM, the higher weight of animals treated with TA3 gives it the importance to investigate further.
Furthermore, this study combines fatty acid research and tumors which is novel and of interest.
However, there are some major concerns around interpretation of results and some minors concerning text and presentation. Here you may find my concerns:
---
Introduction is missing crucial info about the main proteins studied for understanding the paper.
---
Methods are missing catalogue numbers and producers of antibodies, kits and siRNAs, as well as PCR conditions, bioinformatic analysis description and westen blot densitometry analyisis.
---
Figures, as well as Results would benefit from a more detailed description of methods. Results would benefit from small introduction or simple explanations why experiments were done.
---
More specific points in Results section:
It is unclear how the authors chose the 5uM concetration for microarray experiments, what the table 1 presents, how did authors choose pathways to study further (metabolic), why the SREBP-1 was chosen for further analysis, why authors claim change in HMGCS-1 and INSIG-1 mRNAs while qPCR data remain non significant, why authors used 12h or 24h interchangeably for different experiments, why authors claim change in FASN and phospho-ACC when significance was observed only in 1 cell line, why they sometimes use mature/precursor and sometimes nuclear/cytosolic, why they claim in Fig. 1d that the mature protein is decreased in all three cell lines, while the figure and densitometry show differently.
Furthermore, Fig 2A overlaps with Fig 1 A and D and it is unclear why the data repeats.
It is unclear how authors chose proteins which were investigated in detail.
Some blots seem saturated so the densitometry can come in question.
Some images lack densitometry and quantification, like Fig s1 A and B, Fig 2C.
Unclear about what upper part of fig2e means, unclear photo in lower part of 2F.
Fig 2G. Missing explanation on why authors checked phospho protein?
What do the arrows in 2C IF images point to?
Figure 3 A and B show different percentages which differ from text.
Conclusions based on Fig 3G and H are unconvincing. Why check expression of caspases per se?
Invalid statement: "Our results showed that TA3 treatment blocked Akt/Gsk3β signaling-dependent decrease in SREBP-1 maturation, whereas no changes were observed in the phosphorylation of mTOR (Fig. 4a and b)." These results didnt investigate SREBP-1 maturation.
Unconvincing results: "In addition, phoshorylation of AMPK was increased following TA3 (5 µM) treatment" Invalid: "consequently SREBP-1 maturation was reduced."
Fig 4 C and D densitometry: fold of what? Why the controls are not 1? Is the C phosphorilated form divided by the total? If so, this should be corrected throughout the article. Fig 4 D first two lanes are not convincing. Unconvincing: " Interestingly, we found that cleaved PARP and 347
p21 protein levels were slightly decreased in insulin-treated BxPC-3 cells, "
Quantification of Fig 5 C is unconvincing concerning volumes of GEM vs vehicle treated. It seems as though the volumes of GEM-treated tumors are bigger then vehicle.
Rationale behind experiment for Fig. s2 is lacking. What is the purpose of combinational MTT assay here in in vivo results? Maybe it is better to move these data elsewhere in the manuscript. for example, in figure 1.
Fig 5F. Were total or cleaved PARP and cleaved Caspase-3 checked?
Maybe transfer Table 3 to M&M or Supp?
What does Fig 6A left part represent? Are all the data presented in the scheme results? If not, please remove or transfer to supplement with a clear figure legend.
Incorrect statement: "death did not appear even after 24 hours treated with TA3 (Fig. s4)." Please define better. Cell death is obvious for the concentration of 10 and 20.
Interpretations for Fig 5B are unconvincing. Quantification should be done. Incorrect conclusion: "These results demonstrated that the inhibition of SREBP-1 in the TA3-treated subcutaneous BxPC-3 tumor reduced fatty acid synthesis." Conclusions for subcutaneous BxPC-3 tumor can not be made based on cell culture data. Furthermore, since 5 uM TA3 causes cell death in BxPC3 cells, one should be careful about interpreting lower lipid content (Fig 1A).
--
In discussion references are missing. For example: " previous studies " Which? -Lines 428 and 429. "both genes are involved in lipid metabolism" -Lines 434.
References to figures in discussion would come handy.as well.
Unconvincing statement: " which suggests that the cell growth inhibitory effect of TA3 is dependent on SREBP-1-related lipogenesis. "
Use of insulin in experiments should be further clarified.
--
Title is misleading: transcriptional activity of SREBP-1 as well as the fatty acid biosynthesis haven't been checked per se.
--
Kind regards
Author Response
Feb 11th, 2022
Manuscript ID: pharmaceutics-1564084
Title: Inhibition of SREBP-1 Transcriptional Activity by Timosaponin A3 Induces Fatty Acid Biosynthesis Deficiency and Growth Inhibition in Pancreatic Cancer
Dear Editors,
We would like to thank you and the reviewers of the Pharmaceutics for taking the time to review our manuscript. We have made corrections and clarifications in the manuscript after going over the reviewer’s comments.
In this revised manuscript, we carefully responded to each reviewer’s comments, point by point, as in the Author’s response. We hope the revised manuscript will better meet the requirements of your journal for publication. We thank the editors and the reviewers of the Pharmarceutics once again for the constructive review of our paper.
Sincerely yours,
Hyeung-Jin Jang
Responses to the reviwers’s comments:
Review 2.
- Introduction is missing crucial info about the main proteins studied for understanding the paper.
Our response: Thank you for your comments, we have added the sentences for SREBP-1 background in the introduction section (Line 62-75, page 2).
- Methods are missing catalogue numbers and producers of antibodies, kits and siRNAs, as well as PCR conditions, bioinformatic analysis description and western blot densitometry analysis.
Our response: We have added the catalogue numbers and producers of antibodies, kits and siRNAs. In addition, we have added more detailed descriptions for western blot densitometry analysis and bioinformatic analysis in Method section. However, PCR conditions information has been citated to follow the author's guidelines, concise articles.
- Figures, as well as results would benefit from a more detailed description of methods. Results would benefit from small introduction or simple explanations why experiments were done.
Our response: The detailed description of methods and goal has been described in the text.
- More specific points in results section: it is unclear how the authors chose the 5μM concentration for microarray experiments, what the table 1 presents, how did authors choose pathways to study further (metabolic), why the SREBP-1 was chosen for further analysis,
Our response: We have added more detailed descriptions in introduction (Line 62-75, page 2) and results section as reviewer pointed out (Line 283-284, page 7).
why authors claim change in HMGCS-1 and INSIG-1 mRNAs while qPCR data remain non-significant, why authors used 12h or 24h interchangeably for different experiments,
Our response: We have added more detailed descriptions as reviewer pointed out. To add an explanation, HMGCS-1 and INSIG-1 mRNA levels were increased by TA3 treatment compare control even it is not significantly. We repeated the independent experiment, and although p value less than 0.05 was not satisfied due to deviation, HMGCS1 in TA3 treatment was increased 1.2-fold (AsPC1), 1.3-fold (BxPC3), and 1.6-fold (PANC1) compared with control. In addition, INSIG-1 was increased 1.6-fold (AsPC1), 1.9-fold (BxPC3), and 1.3-fold (PANC1) in the TA3 treatment group. (Line 275-282, page 6-7) The study of cellular processes such as differentiation or response to stimulation genes could be presented different time point each experiments. Also, transcription factors can turn on at different times so we used different time point between stimulation genes and transcription factors.
why authors claim change in FASN and phopho-ACC when significance was observed only in 1 cell line, why they sometimes use mature/precursor and sometimes unclear/cytosolic, why they claim in Fig. 1d that the mature protein is decreased in all three cell lines, while the figure an densitometry show differently.
Our response: We have revised the sentences as reviewer pointed out as follows: We also monitored the protein levels of the SREBP-1 downstream targets, the regulation of FASN and phospho-ACC expression by TA3 was significantly suppressed only BxPC-3 cells compare other cell lines. (Fig. 1e). Taken together, the effect of TA3 on the regulation of SREBP-1 expression, including related genes (HMGCS-1 and INSIG-1) and proteins (FASN and ACC), was greater in BxPC-3 cells compared to other cell lines. (Line 288-291, page 7)
Our response: The SREBP-1 forms are presented as precursor form (125 kDa) in the cytosol and the mature form (68 kDa) in the nucleus by western blots. Although the precursor form of SREBP-1 (125 kDa) is thought to be localized to the endoplasmic reticulum membrane, previously studied noted that precursor form of SREBP-1 form could be shown in the salt nuclear extract in HeLa cells like our results. Also, we can find similar results on the company of anti-body products.
- Furthermore, Fig 2A overlaps with Fig 1A and D and it is unclear why the data repeats.
Our response: We have changed the sentences “we incubated BxPC-3 cells with TA3 (2.5 and 5 µM) for 12 h and performed western blotting to confirm the expression levels of FASN and phospho-ACC, two regulators of fatty acid synthase” to “we incubated BxPC-3 cells with TA3 (2.5 and 5 µM) for 12 h and performed western blotting to confirm whether the expression levels of FASN and phospho-ACC, two regulators of fatty acid synthase are regulated by TA3 dose-dependent manner” throughout the text. (Line 307-309, page 9)
- It is unclear how authors chose proteins which were investigated in detail.
Our response: As reviewer pointed out, we added the sentences and suplementary figure for understanding. (Line 62-75, page 2 and Line 283-284, page 7, Fig s1)
- Some blots seem saturated so the densitometry can come in question.
Our response: Unfortunately, we could not change densitometry data from the specimens. Because in the case of proteins appearing in multiple forms (such as cleaved, mature form) on one membrane, the density had to be increased to reveal weak bands.
- Some images lack densitometry and quantification, like Fig s1 A and B, Fig 2C.
Our response: Unfortunately, we could not increase lack densitometry data from the specimens. Because In the case of a protein that appears in multiple forms on one membrane, it had to be stopped at an appropriate exposure in order to prevent the density of strong bands from becoming too saturated due to an increase in exposure.
- Unclear about what upper part of fig2e means, unclear photo in lower part of 2F.
Our response: As reviewer pointed out, we revised the sentences as follows: To evaluate si-SREBP-1 whether induces knock down of SREBP-1 in BxPC-3 cells, we performed the immunoblotting and immunofluorescence. The results have shown that expression of SREBP-1 protein was silenced by si-SREBP-1 in BxPC-3 cells (Fig. 2e and f). (Line 320-330, page 9)
- Fig 2G. Missing explanation on why authors checked phospho protein?
Our response: We added the sentences for background of target proteins in the introduction section (Line 62-75, page 2) and briefly added them in the results section. (Line 330-332, page 9)
- What do the arrows in 2C IF images point to?
Our response: As on the manuscript, we explained arrows mean in figure legend as follows: “Arrows denote nuclear translocation”. In addition, we has been described it in results as follows: “We also found that treatment of BxPC-3 cells with TA3 suppressed the nuclear translocation of transgenic anti-SREBP-1-REP and was reduced its level in the cytosol (Fig. 2c).” (Line 321-323, page 9)
- Figure 3A and B show different percentages which differ from text.
Our response: Thank you for your comments. Our cell cycle graph has been represented the independent triplicate experiments. Therefore, cell cycle distribution percentage on Fig 3b looks like a different with Fig 3a by average value of triplicate experiments.
- Conclusion based on Fig3G and H are unconvincing. Why check expression of caspases per se?
Our response: We are pleasure to the reviewer’s comments, we looked at the data and found that the cleaved caspase-3 and caspase-3 were swapped in the process of organizing the figures. Thus, we have corrected the expressions in the revised manuscript as reviewer pointed out as follows: We observed the expression of pro-apoptotic genes (Cleaved Caspase-3, -9, and Bid) and anti-apoptotic genes (Bcl-2) by qRT-PCR (Fig. 3g). Consequently, Caspase-9 mRNA level was increased and Bcl-2 mRNA level was decreased in TA3 dose-dependent manner. Both genes were significantly regulated in 5 μM. Whereas, pro-apoptosis caspase-3 and Bid were not regulated in a dose-dependent manner of TA3, and Caspase 3 was elevated at 2.5 μM, and 5 μM of TA3, respectively. Furthermore, both concentrations (2.5 and 5 µM) of TA3 clearly increased the expression of cleaved PARP, widely used in apoptotic markers, while cleaved caspase-3 only slightly increased at 5 μM compare to control (Fig. 3g). (Line365-373, page 11)
- Invalid statement: Our results showed that TA3 treatment blocked Akt/Gsk3b signaling-dependent decrease in SREBP-1 maturation, whereas no changes were observed in the phosphorylation of mTOR (Fig. 4a and b). These results didn’t investigate SREBP-1 maturation.
Our response: As reviewer pointed out, We have revised the manuscript due to misleading expressions in the text.(Line 390-392, page 13)
- Unconvincing results: “in addition, phosphorylation of AMPK was increased following TA3 (5uM) treatment” invalid: “consequently SREBP-1 maturation was reduced”
Our response: As reviewer pointed out, we have deleted the sentence the manuscript due to misleading expressions in the text. (Line 394, page 13)
- Fig 4c and d densitometry: fold of what? Why the controls are not 1? Is the C phosphorylated form divided by the total? If so this should be corrected throughout the article. Fig 4D first two lanes are not convincing. Unconvincing: “interestingly, we found that cleaved PARP and p21 protein levels were slightly decreased in insulin-treated BxPC-3 cells.”
Our response: As reviewer pointed out, we have added the sentences for the how they calculate the phosphorylated form, cleaved form and mature form in immunoblotting method. In addition, we have added the sentences for the understand insulin treatment as the reviewer suggested. (Line 159-161, page 4)
- Quantification of Fig 5C is unconvincing concerning volumes of GEM vs. vehicle treated. It seems as though the volumes of GEM-treated tumors are bigger then vehicle.
Our response: When viewed as a picture in Fig 5c, the GEM tumor looks not much different from the vehicle because, GEM tumor shape is flat. However, when the size is measured, the difference appears as in the result of Fig 5d.
- Rationale behind experiment for Fig s2 is lacking. What is the purpose of combinational MTT assay here in in vivo results? Maybe it is better to move these data elsewhere in the manuscript. For example, in figure 1.
Our response: The combination therapy of gemcitabine and TA3 will be continued in future studies. In addition, tumor growth inhibition of GEM-treated mice and TA3-treated mice was compared in Fig 5. In this regard, the results of Fig s2 has been described as supporting results. Thus, we unfortunately difficult to change to Fig 1.
- Fig 5F. Were total or cleaved PARP and cleaved Cas3 checked?
Our response: We used cleaved caspase 3 and total PARP in IHC assay as we mentioned in the materials and method section. We have revised “caspase 3” to “cleaved caspase 3” in the figure 5f and figure legend.
- Maybe transfer table 3 to M&M or supp?
Our response: Thank you for your comments. However, this is the main result of quantitative and qualitative analysis of fatty acids, which is helpful in understanding the quantitative analysis of fatty acids in tissues. (Table 3, Line 460-463, page 16)
- What does fig 6a left part represent? Are all the data presented in the scheme results? If not, please remove or transfer to supplement with a clear figure legend.
Our response: As reviewer pointed out, we have deleted the figure legend and remove schematic in manuscript. (Figure 6a, page 17)
- Incorrect statement: “death did not appear even after 24 hours treated with TA3 (fig. s4)” please define better. Cell death is obvious for the concentration of 10 and 20.
Our response: “death did not appear even after 24 hours treated with TA3 (fig. s4)” sentences have reviesed as the reviewer pointed out. (Figure s4 revised to Figure s5, Line474-475, page 17)
- Interpretations for fig 5b are unconvincing. Quantification should be done. Incorrect conclusion: “these results demonstrated that the inhibition of srebp-1 in the TA3-treated subcutaneous BxPC-1 tumor reduced fatty acid synthesis” conclusions for subcutaneous BxPC-3 tumor cannot be made based on cell culture data. Furthermore, since 5uM TA3 causes cell death in BxPC3 cells, one should be careful about interpreting lower lipid content (Fig 1A)
Our response: We have added the sentences for “preparation of sample and fatty acid standards” in method section. We extracted the metabolites from all mouse tumors with equal weight (5 mg/ml) and quantified them. (Line 224-233, page 5-6)
- In discussion references are missing. “previous studies” which? -lines 428 and 429. “Both genes are involved in lipid metabolism”-lines 434. References to figures in discussion would come handy. As well.
Our response: We follow the reviewer’s suggestion. (Line 492 and 497, page 17 and 18)
- Unconvincing statement: “which suggests that the cell growth inhibitory effect of TA3 is dependent on SREBP-1-related lipogenesis” Use of insulin in experiments should be further clarified.
Our response: There is some reason in what the reviewer pointed out. Therefore, we added more detailed description in main text for understand our hypothesis and results.
- Title is misleading: transcriptional activity of SREBP-1 as well as the fatty acid biosynthesis haven’t been checked per se.
Our response: There is some reason in what the reviewer pointed out. However, we confirmed a decrease in fatty acid synthase activity such as FASN and ACC through TA3-induced inhibition of SREBP-1 in BxPC-3 cells. Moreover, we suggested reduced levels of saturated fatty acids such as palmitate and malate in tumor tissue due to TA3 treatment.

Reviewer 3 Report
In the current manuscript, the authors Kim et al. tried to evaluate effect of Timosaponin A3 in fatty acid metabolism and in pancreatic cancer. There are several, some very critical, concerns which should be addressed.
- Line 37: The numbers of new cancer cases and deaths are each year estimated. There is a very new publication for 2022, the five-year survival rate is now at 11% (PMID: 35020204).
- Line 42: Are GEM and 5-FU still standard options? How about FOLFIRINOX? Please include recent publication or guideline.
- Line 70 (Methods): Please include methods for western blotting.
- Line 237 (Figure 1d): “The expression level of the mature form of SREBP-1 was decreased in all three cell lines following TA3 treatment” The data is not convincing. Please perform the experiment again and re-analyze the data. To be sure that cellular fractionation worked, please include nuclear and cytosol markers in your western blotting as well. Also, please label the size of your bands (kDa) (for all western blot experiments in the manuscript).
- Line 240 (Figure 1e): Did author analyze ACC1 or ACC2? Which phosphorylation site did the author analyze (presumably S79, then ACC1?)? AMPK phosphorylates and directly inhibits ACCs. Somewhat I do not understand why the authors try to show less FASN expression and less P-ACC.
- Line 261 (Figure 2a): ACC phosphorylation results in its inhibition leading to suppression of lipogenesis. Here I do not understand why the authors try to show FASN downregulation and less ACC phosphorylation.
- Line 275 (Figure 2d): The authors labeled pSREBP-1 was from the nucleus, in figure 1d the authors labeled pSREBP-1 was from cytosol.
- Line 280 (Figure 2g): same to the points 6 and 7. The authors showed less ACC phosphorylation after siRNA.
- Line 280 (Figure 2h): While western blot data for p27 and p21 are convincing, blotting data for PARP is not convincing.
- Line 332 (Figure 4a): Blotting data for p-Akt is convincing, but the other blots are not convincing.
- Line 341 (Figure 4d): same to the point 7, pSREBP-1 in the nucleus?
- Line 347 (Figure 4e): same to the point 6, less FASN and less P-ACC?
- Line 363 (Figure 5a): the schema shows that “BxPC3 injection at D-7, randomization D-0, treatment stop D-20, and at D-25 sacrifice”. In the method section, the authors described that “euthanized 1 week after the end of treatment”. What exactly did the author perform the experiment? If the method part is correct, please change the scheme Figure 5a. Further, how the authors know “the tumors reached a diameter of 0.5 cm”? Does it take exactly 7 day as in Figure 5a? How did the authors exactly know tumor volume between days 2 to 18?
- Line 386 (Figure 5f): Please include statistical analysis in the figure. Is it IHC for cleaved caspase 3?
Author Response
Feb 11th, 2022
Manuscript ID: pharmaceutics-1564084
Title: Inhibition of SREBP-1 Transcriptional Activity by Timosaponin A3 Induces Fatty Acid Biosynthesis Deficiency and Growth Inhibition in Pancreatic Cancer
Dear Editors,
We would like to thank you and the reviewers of the Pharmaceutics for taking the time to review our manuscript. We have made corrections and clarifications in the manuscript after going over the reviewer’s comments.
In this revised manuscript, we carefully responded to each reviewer’s comments, point by point, as in the Author’s response. We hope the revised manuscript will better meet the requirements of your journal for publication. We thank the editors and the reviewers of the Pharmarceutics once again for the constructive review of our paper.
Sincerely yours,
Hyeung-Jin Jang
Responses to the reviwers’s comments:
Review 3.
- Line 37: the numbers of new cancer cases and deaths are each year estimated. There is a very new publication for 2022, the five-year survival rate is now at 11% (PMID: 35020204).
Our response: Thank you for your comments. We follow the reviewer suggestion. (Line 37, page 1)
- Line 42: Are GEM and 5-FU still standard options? How about FOLFIRINOX? Please include recent publication or guideline.
Our response: As reviewers have noted, FOLFIRINOX is also being used as a treatment. In this study, GEM was mentioned using GEM as a standard material, and GEM was selected as a standard material because it is a relatively inexpensive drug and is generally used in many papers. We added your comments in introduction section. (Line 43, page 1)
- Line 70 (Methods): please include methods for western blotting.
Our response: The method for western blotting commented by the reviewer is written in section 2.7 cell lysis and immunoblotting in the original manuscript. (Line 148-161, page 4)
- Line 237 (Figure 1d): “The expression level of the mature form of SREBP-1 was decreased in all three cell lines following TA3 treatment” The data is not convincing. Please perform the experiment again and re-analyze the data. To be sure that cellular fractionation worked. Please include nuclear and cytosol markers in your western blotting as well. Also, please label the size of your bands (kDa)
Our response: We agree with the reviewer’s opinion. Unfortunately, the author who took the lead in this experiment has been promoted and is in a different place, so it is difficult to proceed with the experiment again. The goal of this results is to find the connection between SREBP-1 and TA3 in pancreatic cancer cells and which cell is most regulated by TA3. We suggest that SREBP-1 expression was decreased 0.9-fold (AsPC-1), 0.6-fold (BxPC3), and 0.8-fold (PANC-1) by TA3 in figure 1d. We have revised sentences in the revised manuscript as reviewer pointed out. (Line 286-288, page 7)
- Line 240 (Figure 1e): did author analyze ACC1 or ACC2? Which phosphorylation site did the author analyze (presumably s79, then ACC1?)? AMPK phosphorylates and directly inhibits ACCs. Some what I do not understand why the authors try to show less FASN expression and less P-ACC.
Our response: We have added the target proteins information in introduction section for understand. (Line 66-72, page 2) In addition, we have revised the sentences in results as reviewer pointed out. (Line 288-291, page 7)
- Line 261 (Figure 2a): ACC phosphorylation results in its inhibition leading to suppression of lipogenesis. Here I do not understand why the authors try to show FASN downregulation and less ACC phosphorylation.
Our response: We have added the information for target proteins as reviewer pointed out. (Line 66-72, page 2) In addition, we have changed the sentences “we incubated BxPC-3 cells with TA3 (2.5 and 5 µM) for 12 h and performed western blotting to confirm the expression levels of FASN and phospho-ACC, two regulators of fatty acid synthase” to “we incubated BxPC-3 cells with TA3 (2.5 and 5 µM) for 12 h and performed western blotting to confirm whether the expression levels of FASN and phospho-ACC, two regulators of fatty acid synthase are regulated by TA3 dose-dependent manner” throughout the text. (Line 307-309, page 9)
- Line 275 (Figure 2d): The authors labeled pSREBP-1 was from the nucleus, in figure 1d the authors labeled pSREBP-1 was form cytosol.
Our response: The SREBP-1 forms are presented as precursor form (125kDa) in the cytosol and the mature form (68kDa) in the nucleus by western blots. Although the precursor form of SREBP-1 (125kDa) is thought to be localized to the endoplasmic reticulum membrane, previously studied (Wang et al. Cell 1994;77(1):53-62, Wu et al. Am J Physiol 1999;277(6):e1087-94) noted that precursor form of SREBP-1 form could be shown in the salt nuclear extract in HeLa cells like our results. Also, we can find similar results on the company of anti-body products.
- Line 280 (Figure 2g): same to the points 6 and 7. The authors showed less ACC phosphorylation after siRNA.
Our response: According to the reviewer's comments, we have added information about the target protein to the introduction in the same way as in answers 6 and 7. (Line 62-75, page 2) Also, we have revised the sentences in the results section (Line330-334, page 9) as reviewer pointed out.
- Line 280 (Figure 2h): while western blot data for p27 and p21 are convincing, blotting data for PARP is not convincing.
Our response: We have revised the sentences and replaced blotting data in the revised manuscript as reviewer pointed out as follows: Silencing of SREBP-1 significantly inhibited lipid metabolism-related enzymes, and in-creased the expression of cleaved PARP (1.4-fold) and p21 (1.9-fold) compared to control (Fig. 2g and h). Although no significant results were obtained due to high deviation, the expression of p27 was greatly increased 2.9-fold by si-SREBP-1. (Line 330-334, page 9)
- Line 332 (Figure 4a): Blotting data for p-Akt is convincing, but the other blots are not convincing.
Our response: We have replaced blotting data as reviewer pointed out. In addition, we have been mentioned “whereas no changes were observed in the phosphorylation of mTOR (Fig. 4a and b).” in results section. And we have revised the sentences as reviewer pointed out as follow: “phosphorylation of AMPK was slightly increased following TA3 (5 µM) treatment.” (Line 390-394, page 13)
- Line 341 (Figure 4d): Same to the point 7, pSREBP-1 in the nucleus?
Our response: Same as answer point 7, the SREBP-1 forms are presented as precursor form (125 kDa) in the cytosol and the mature form (68 kDa) in the nucleus by western blots. Although the precursor form of SREBP-1 (125 kDa) is thought to be localized to the endoplasmic reticulum membrane, previously studied (Wang et al. Cell 1994;77(1):53-62, Wu et al. Am J Physiol 1999;277(6):e1087-94) noted that precursor form of SREBP-1 form could be shown in the salt nuclear extract in HeLa cells like our results. Also, we can find similar results on the company of anti-body products.
- Line 347 (Figure 4e): Same to the point 6, less FASN and less p-ACC?
Our response: We have added the target proteins information in the introduction section for understand. (Line 62-75, page 2)
- Line 363 (Figure 5a): the schema shows that “BxPC3 injection at D-7, randomization D-0, treatment stop D-20, and at D-25 sacrifice”. In the method section, the authors described that “euthanized 1 week after the end of treatment”. What exactly did the author perform the experiment? If the method part is correct, please change the scheme Figure 5a. Further, how the authors know “the tumors reached a diameter of 0.5 cm"? Does it take exactly 7 day as in Figure 5a? How did the authors exactly know tumor volume between days 2 to 18?
Our response: Thank you for your comments, we have revised 5 days in the method section. (Line 205, page 5) As on the original manuscript, we explained the how tumor volume was measured in the method section (Line 204-207, page 5).
- Line 386 (Figure 5f): Please include statistical analysis in the figure. Is it IHC for cleaved caspase 3?
Our response: Sorry for any inconvenience. We used cleaved caspase 3 in IHC assay as we mentioned in the materials and method section. We have revised “caspase 3” to “cleaved caspase 3” in the figure 5f and figure legend. IHC staining showed a clear difference in protein expression level between vehicle and TA3 slides even without statistical analysis.

Round 2
Reviewer 1 Report
NA
Author Response
Thank you.

Reviewer 2 Report
Dear Authors, thank you for your timely review.
Here I provide my further comments coloured with violet and yellow for the ease of reading. The text is now clearer. But further clarifications are needed text-wise. I hope this comments find you well.
Kind regards
Responses to the authors’ responses:
- Introduction is missing crucial info about the main proteins studied for understanding the paper.
Our response: Thank you for your comments, we have added the sentences for SREBP-1 background in the introduction section (Line 62-75, page 2).
REVIEWER: Thank you.
Also, it is not introduced how INSIG1 and HMGCS1 are connected to SREBP-1.
Please check the term " multiple drug " in the added sentences. What does it mean? Please clarify or use different term.
Line 74, 75: Please clarify how can SREBP-1 be a predictor of poor prognosis in patients if its absence leads to inhibition of proliferation and induction of apoptosis?
- Methods are missing catalogue numbers and producers of antibodies, kits and siRNAs, as well as PCR conditions, bioinformatic analysis description and western blot densitometry analysis.
Our response: We have added the catalogue numbers and producers of antibodies, kits and siRNAs. In addition, we have added more detailed descriptions for western blot densitometry analysis and bioinformatic analysis in Method section. However, PCR conditions information has been citated to follow the author's guidelines, concise articles.
REVIEWER: Thank you.
PCR cycling conditions (temperature and cycle lenght) haven't been described in the paper that was cited. Please insert.
- Figures, as well as results would benefit from a more detailed description of methods. Results would benefit from small introduction or simple explanations why experiments were done.
Our response: The detailed description of methods and goal has been described in the text.
REVIEWER: Thank you.
Figures would benefit from a more detailed description of methods, e.g. Fig 1B, D, E lacks crucial information about the method used.
Furthermore, part of title for Fig1: " and inhibits SREBP-1 transcriptional activity in PC cells. " is not correct, since transcriptional activity wasn't measured by any functional assay. Please soften the statement.
- More specific points in results section: it is unclear how the authors chose the 5μM concentration for microarray experiments, what the table 1 presents, how did authors choose pathways to study further (metabolic), why the SREBP-1 was chosen for further analysis,
Our response: We have added more detailed descriptions in introduction (Line 62-75, page 2) and results section as reviewer pointed out (Line 283-284, page 7).
REVIEWER: Thank you.
It is unclear how the authors chose the 5μM concentration for microarray experiments.
Table2 legend is lacking, it is unclear what it represents when compared to Fig1B.
Fig 1B) Maybe change" The number of gene transcripts in each category was listed." to "X-axis represent the number of gene transcripts in each category. Negative values represent downregulated and positive upregulated genes after TA3 treatment." or similar.
Line 273: what is the meaning of " functional genes "? Please clarify or use different term.
Line 276: double!
Additional: HMGCR results haven't been mentioned in the text.
What does it mean: " In addition, we found that two genes were interacted with 20 genes, which was related with SREBP signaling pathway using GeneMANIA (Fig. s1)."? I guess this is the explanation to my question: Why the SREBP-1 was chosen for further analysis? But it is unclear. Please clarify or use different term.
Line 284: Please write here why you have decided to check nuclear and cytoplasmic fractions of SREBP-1. Insert more details with reference.
Line 289: missing reference: " SREBP-1 downstream targets "
why authors claim change in HMGCS-1 and INSIG-1 mRNAs while qPCR data remain non-significant, why authors used 12h or 24h interchangeably for different experiments,
Our response: We have added more detailed descriptions as reviewer pointed out. To add an explanation, HMGCS-1 and INSIG-1 mRNA levels were increased by TA3 treatment compare control even it is not significantly. We repeated the independent experiment, and although p value less than 0.05 was not satisfied due to deviation, HMGCS1 in TA3 treatment was increased 1.2-fold (AsPC1), 1.3-fold (BxPC3), and 1.6-fold (PANC1) compared with control. In addition, INSIG-1 was increased 1.6-fold (AsPC1), 1.9-fold (BxPC3), and 1.3-fold (PANC1) in the TA3 treatment group. (Line 275-282, page 6-7) The study of cellular processes such as differentiation or response to stimulation genes could be presented different time point each experiments. Also, transcription factors can turn on at different times so we used different time point between stimulation genes and transcription factors.
REVIEWER: Thank you. The significance of the results without statistical significance is doubtful, but I can agree that there is a similar trend of change throughout the data.
why authors claim change in FASN and phopho-ACC when significance was observed only in 1 cell line, why they sometimes use mature/precursor and sometimes unclear/cytosolic, why they claim in Fig. 1d that the mature protein is decreased in all three cell lines, while the figure an densitometry show differently.
Our response: We have revised the sentences as reviewer pointed out as follows: We also monitored the protein levels of the SREBP-1 downstream targets, the regulation of FASN and phospho-ACC expression by TA3 was significantly suppressed only BxPC-3 cells compare other cell lines. (Fig. 1e). Taken together, the effect of TA3 on the regulation of SREBP-1 expression, including related genes (HMGCS-1 and INSIG-1) and proteins (FASN and ACC), was greater in BxPC-3 cells compared to other cell lines. (Line 288-291, page 7)
REVIEWER: Thank you.
Our response: The SREBP-1 forms are presented as precursor form (125 kDa) in the cytosol and the mature form (68 kDa) in the nucleus by western blots. Although the precursor form of SREBP-1 (125 kDa) is thought to be localized to the endoplasmic reticulum membrane, previously studied noted that precursor form of SREBP-1 form could be shown in the salt nuclear extract in HeLa cells like our results. Also, we can find similar results on the company of anti-body products.
REVIEWER: Thank you.
- Furthermore, Fig 2A overlaps with Fig 1A and D and it is unclear why the data repeats.
Our response: We have changed the sentences “we incubated BxPC-3 cells with TA3 (2.5 and 5 µM) for 12 h and performed western blotting to confirm the expression levels of FASN and phospho-ACC, two regulators of fatty acid synthase” to “we incubated BxPC-3 cells with TA3 (2.5 and 5 µM) for 12 h and performed western blotting to confirm whether the expression levels of FASN and phospho-ACC, two regulators of fatty acid synthase are regulated by TA3 dose-dependent manner” throughout the text. (Line 307-309, page 9)
REVIEWER: Thank you.
- It is unclear how authors chose proteins which were investigated in detail.
Our response: As reviewer pointed out, we added the sentences and suplementary figure for understanding. (Line 62-75, page 2 and Line 283-284, page 7, Fig s1)
REVIEWER: Thank you.
- Some blots seem saturated so the densitometry can come in question.
Our response: Unfortunately, we could not change densitometry data from the specimens. Because in the case of proteins appearing in multiple forms (such as cleaved, mature form) on one membrane, the density had to be increased to reveal weak bands.
REVIEWER: Thank you. While I can understand the difficulty of filming multiple forms with different abundance, the solution here is to film them separately with different times. I understand that this is not possible at the moment.
- Some images lack densitometry and aication, like Fig s1 A and B, Fig 2C.
Our response: Unfortunately, we could not increase lack densitometry data from the specimens. Because In the case of a protein that appears in multiple forms on one membrane, it had to be stopped at an appropriate exposure in order to prevent the density of strong bands from becoming too saturated due to an increase in exposure.
REVIEWER: Thank you. Densitometry is done post-filming. Please analyse Fig s2 A and quantify S2B and Fig2C.
- Unclear about what upper part of fig2e means, unclear photo in lower part of 2F.
Our response: As reviewer pointed out, we revised the sentences as follows: To evaluate si-SREBP-1 whether induces knock down of SREBP-1 in BxPC-3 cells, we performed the immunoblotting and immunofluorescence. The results have shown that expression of SREBP-1 protein was silenced by si-SREBP-1 in BxPC-3 cells (Fig. 2e and f). (Line 320-330, page 9)
REVIEWER: It is still unclear what the upper part of fig2e means. Please explain what does the asterix mean in fig2e? and what the ns refers to. Also, comment this in the text.
ADDITIONAL: Fig 2g: why wasn't pACC calculated relative to ACC in this case?
- Fig 2G. Missing explanation on why authors checked phospho protein?
Our response: We added the sentences for background of target proteins in the introduction section (Line 62-75, page 2) and briefly added them in the results section. (Line 330-332, page 9)
REVIEWER: Missing explanation on why authors checked phosphorilated form of the protein ACC? Please insert relevant reference.
Line 330: please change inhibited to "lowered the expression" or similar.
Line 331: Please change " increased the expression " to increased"
- What do the arrows in 2C IF images point to?
Our response: As on the manuscript, we explained arrows mean in figure legend as follows: “Arrows denote nuclear translocation”. In addition, we has been described it in results as follows: “We also found that treatment of BxPC-3 cells with TA3 suppressed the nuclear translocation of transgenic anti-SREBP-1-REP and was reduced its level in the cytosol (Fig. 2c).” (Line 321-323, page 9)
REVIEWER: Thank you. I see now. Please explain what is "transgenic anti-SREBP-1-REP"?
- Figure 3A and B show different percentages which differ from text.
Our response: Thank you for your comments. Our cell cycle graph has been represented the independent triplicate experiments. Therefore, cell cycle distribution percentage on Fig 3b looks like a different with Fig 3a by average value of triplicate experiments.
REVIEWER: Thank you. This information should be added in the description of 3a.
- Conclusion based on Fig3G and H are unconvincing. Why check expression of caspases per se?
Our response: We are pleasure to the reviewer’s comments, we looked at the data and found that the cleaved caspase-3 and caspase-3 were swapped in the process of organizing the figures. Thus, we have corrected the expressions in the revised manuscript as reviewer pointed out as follows: We observed the expression of pro-apoptotic genes (Cleaved Caspase-3, -9, and Bid) and anti-apoptotic genes (Bcl-2) by qRT-PCR (Fig. 3g). Consequently, Caspase-9 mRNA level was increased and Bcl-2 mRNA level was decreased in TA3 dose-dependent manner. Both genes were significantly regulated in 5 μM. Whereas, pro-apoptosis caspase-3 and Bid were not regulated in a dose-dependent manner of TA3, and Caspase 3 was elevated at 2.5 μM, and 5 μM of TA3, respectively. Furthermore, both concentrations (2.5 and 5 µM) of TA3 clearly increased the expression of cleaved PARP, widely used in apoptotic markers, while cleaved caspase-3 only slightly increased at 5 μM compare to control (Fig. 3g). (Line365-373, page 11)
REVIEWER: Thank you.
- Invalid statement: Our results showed that TA3 treatment blocked Akt/Gsk3b signaling-dependent decrease in SREBP-1 maturation, whereas no changes were observed in the phosphorylation of mTOR (Fig. 4a and b). These results didn’t investigate SREBP-1 maturation.
Our response: As reviewer pointed out, We have revised the manuscript due to misleading expressions in the text.(Line 390-392, page 13)
REVIEWER: Thank you. Added part of the sentence: "SREBP-1 inhibition was induced" is unclear and added " GSK3b were inhibited " is incorrect, since data show increase of phosphoGSK3b. Please clarify or use different term.
- Unconvincing results: “in addition, phosphorylation of AMPK was increased following TA3 (5uM) treatment” invalid: “consequently SREBP-1 maturation was reduced”
Our response: As reviewer pointed out, we have deleted the sentence the manuscript due to misleading expressions in the text. (Line 394, page 13)
REVIEWER: Thank you.
- Fig 4c and d densitometry: fold of what? Why the controls are not 1? Is the C phosphorylated form divided by the total? If so this should be corrected throughout the article. Fig 4D first two lanes are not convincing. Unconvincing: “interestingly, we found that cleaved PARP and p21 protein levels were slightly decreased in insulin-treated BxPC-3 cells.”
Our response: As reviewer pointed out, we have added the sentences for the how they calculate the phosphorylated form, cleaved form and mature form in immunoblotting method. In addition, we have added the sentences for the understand insulin treatment as the reviewer suggested. (Line 159-161, page 4)
REVIEWER: Thank you.
- Quantification of Fig 5C is unconvincing concerning volumes of GEM vs. vehicle treated. It seems as though the volumes of GEM-treated tumors are bigger then vehicle.
Our response: When viewed as a picture in Fig 5c, the GEM tumor looks not much different from the vehicle because, GEM tumor shape is flat. However, when the size is measured, the difference appears as in the result of Fig 5d.
REVIEWER: Thank you. This should be written in the text describing results in Line 433 after " compared to the vehicle group (Fig. 5c and d)."
- Rationale behind experiment for Fig s2 is lacking. What is the purpose of combinational MTT assay here in in vivo results? Maybe it is better to move these data elsewhere in the manuscript. For example, in figure 1.
Our response: The combination therapy of gemcitabine and TA3 will be continued in future studies. In addition, tumor growth inhibition of GEM-treated mice and TA3-treated mice was compared in Fig 5. In this regard, the results of Fig s2 has been described as supporting results. Thus, we unfortunately difficult to change to Fig 1.
REVIEWER: Thank you. Ok.
- Fig 5F. Were total or cleaved PARP and cleaved Cas3 checked?
Our response: We used cleaved caspase 3 and total PARP in IHC assay as we mentioned in the materials and method section. We have revised “caspase 3” to “cleaved caspase 3” in the figure 5f and figure legend.
REVIEWER: Thank you.
- Maybe transfer table 3 to M&M or supp?
Our response: Thank you for your comments. However, this is the main result of quantitative and qualitative analysis of fatty acids, which is helpful in understanding the quantitative analysis of fatty acids in tissues. (Table 3, Line 460-463, page 16)
REVIEWER: This result still feels unnecessary in the main results section since it is a means to obtain results, and not the main result. But ok.
- What does fig 6a left part represent? Are all the data presented in the scheme results? If not, please remove or transfer to supplement with a clear figure legend.
Our response: As reviewer pointed out, we have deleted the figure legend and remove schematic in manuscript. (Figure 6a, page 17)
REVIEWER: Thank you.
- Incorrect statement: “death did not appear even after 24 hours treated with TA3 (fig. s4)” please define better. Cell death is obvious for the concentration of 10 and 20.
Our response: “death did not appear even after 24 hours treated with TA3 (fig. s4)” sentences have reviesed as the reviewer pointed out. (Figure s4 revised to Figure s5, Line474-475, page 17)
REVIEWER: Thank you.
- Interpretations for fig 5b are unconvincing. Quantification should be done. Incorrect conclusion: “these results demonstrated that the inhibition of srebp-1 in the TA3-treated subcutaneous BxPC-1 tumor reduced fatty acid synthesis” conclusions for subcutaneous BxPC-3 tumor cannot be made based on cell culture data. Furthermore, since 5uM TA3 causes cell death in BxPC3 cells, one should be careful about interpreting lower lipid content (Fig 1A)
Our response: We have added the sentences for “preparation of sample and fatty acid standards” in method section. We extracted the metabolites from all mouse tumors with equal weight (5 mg/ml) and quantified them. (Line 224-233, page 5-6)
REVIEWER: Thank you.
Incorrect conclusion here, please revise: “these results demonstrated that the inhibition of srebp-1 in the TA3-treated subcutaneous BxPC-1 tumor reduced fatty acid synthesis” Conclusions about fatty acids for subcutaneous BxPC-3 tumor cannot be made based on cell culture data! furthermore, synthesis of fatty acids haven't been checked. ORO staining only can checks abundance or content, but not synthesis!
Fig 3 d doesn't have a legend.
Line 481: Lipogenesis hasn't been checked, please revise this statement to something softer (probable, possible or suggesting inhibition): "through inhibition of lipogenesis."
Fig 6 title is missleading on "synthesis" since synthesis hasn't been checked: " TA3 inhibitory effects against fatty acid synthesis in pancreatic tumor tissues. " Please change to "TA3 lowers fatty acid content in pancreatic tumor " or similar.
- In discussion references are missing. “previous studies” which? -lines 428 and 429. “Both genes are involved in lipid metabolism”-lines 434. References to figures in discussion would come handy. As well.
Our response: We follow the reviewer’s suggestion. (Line 492 and 497, page 17 and 18)
REVIEWER: Thank you. References to figures in discussion would ease the reading of the paper. For example in the line 516 or Line 575 after: " the in vitro results. "
- Unconvincing statement: “which suggests that the cell growth inhibitory effect of TA3 is dependent on SREBP-1-related lipogenesis” Use of insulin in experiments should be further clarified.
Our response: There is some reason in what the reviewer pointed out. Therefore, we added more detailed description in main text for understand our hypothesis and results.
REVIEWER: Line 531: “related lipogenesis” The results that came before this sentence don't discuss about lipids so this conclusion is incorrect. Also, lipogenesis hasn't been tested.
- Title is misleading: transcriptional activity of SREBP-1 as well as the fatty acid biosynthesis haven’t been checked per se.
Our response: There is some reason in what the reviewer pointed out. However, we confirmed a decrease in fatty acid synthase activity such as FASN and ACC through TA3-induced inhibition of SREBP-1 in BxPC-3 cells. Moreover, we suggested reduced levels of saturated fatty acids such as palmitate and malate in tumor tissue due to TA3 treatment.
REVIEWER: The paper has given indirect proofs that the fatty acid synthesis is deregulated and that the SREBP-1's transcriptional activity has been hindered. But the paper doesn't provide direct functional experiments which checked
-transcriptional activity of SREBP-1 nor
-fatty acid biosynthesis
and thus it would be of benefit that the title softens a bit.
Related to this:
-Please, see if you may remove transcriptional activity from figure legend Fig1 to something more suitable like: " TA3 regulates lipid metabolism, lowers the nuclear localization of SREBP-1 and the expression of its target genes in PC cells. "
-Please, see if you may remove transcriptional activity from figure legend Fig1 to something more suitable like: " TA3 regulates lipid metabolism and inhibits SREBP-1 and its target genes in PC cells. "
-Please, see if you may remove transcriptional activity from figure legend Fig2 to something more suitable like: " TA3 reduces mature SREBP-1 in BxPC-3 cells "
Line 477: Please change if possible : "These results demonstrated that the inhibition of SREBP-1 in the TA3- " to: "These results implicate" or "These results suggest" or similar..
figure 6. legend: Please omit (delete) "synthesis " if possible: " TA3 inhibitory effects against fatty acid synthesis in pancreatic tumor tissues.
Additional remarks:
-Line 521: Please correct the reference: " Maria T. and Johan"
-Line 345: correct this: " expression of SREBP-1 nuclear translocation " to "expression of nuclear SREBP-1" or similar.
Fig 5.A text "as described in Materials and Methods." is inappropriate. figure has to be "stand alone" and all the important info has to be written, without references to other parts of the paper.
Line 507: Please check if "overexpression" here is correct or add details or refenrence to figure to make it clearer.
Line 514: Please check if "on the nucleic" is correct.
Line 516, 517: "reduction..is induced" Please change to a more suitable wording.
Line 553: " inhibitory effect of Akt and GSK3β activity by TA3" is incorrect for GSK3B, since your data show that TA3 activates GSK3B! Same comment for Line 583:" TA3 reduced Akt/GSK3β signaling "
Line 560: missing reference
Line 416 and 388: I wonder why was the conclusion made that the inhibition of SREBP-1 TA3 appears independent of the Akt/Gsk3β mechanism? Please add explanation.
Line 565: change " enzymes " and " factors " to singular if correct.
Thank you.
Author Response
Hyeung Jin Jang, Ph.D.
Professor
Department of Biochemistry,
College of Korean Medicine
Kyung Hee University,
26, KungHeedae-ro, Dongdaemun-gu,
Seoul, 02447, Republic of Korea
E-mail: [email protected]
Apr 18th, 2022
Manuscript ID: Pharmaceutics-1564084
Revised tittle:
Timosaponin A3 Inhibits Palmitate and Stearate through Suppression of SREBP-1 in Pancreatic cancer
Dear Editors,
Thanks to the reviewers and Pharmaceutics for taking the time to review the article. Some additional experiments, corrections, and clarifications were made to the manuscript after receiving comments from the judges.
In the revised manuscript, each reviewer's comments were carefully answered one by one, just like the authors' responses. We hope that the revised manuscript better meets the journal's requirements for publication. We would like to thank the editors and reviewers of Pharmarceutics once again for a constructive review of our paper.
Sincerely yours,
Hyeung-Jin Jang
Responses to the authors’ responses:
Reviewer #2
- Introduction is missing crucial info about the main proteins studied for understanding the paper.
Our response: Thank you for your comments, we have added the sentences for SREBP-1 background in the introduction section (Line 62-75, page 2).
REVIEWER:Thank you.
REVIEWER:Also, it is not introduced how INSIG1 and HMGCS1 are connected to SREBP-1.
Our response: Thank you for your comments. We mentioned that in the discussion section (Line 546-549). Nevertheless, based on your comments, we added sentences in the introduction section: (Line 65-73) SREBP, as a transcription factor, is bound by insulin-induced gene 1 (INSIG-1) and the SREBP cleavage-activating protein (SCAP)/SREBP complex remains an inactive pre-cursor in the ER (endoplasmic reticulum). SREBP is a transcription factor, an inactive precursor that binds to the ER (endoplasmic reticulum) and . SREBPnuclear envelope, is activated by proteolytic cleavage of the Golgi apparatus to produce an N-terminal mature transcription factor that translocates to the nucleus.” and “SREBP-2 is involved in cholestrol synthesis pathway and activate expression of genes such as HMG-CoA reductase (HMGCR), HMG-CoA synthease (HMGCS) and LDL re-ceptor (LDLR).
REVIEWER:Please check the term " multiple drug " in the added sentences. What does it mean? Please clarify or use different term.
Our response: Multiple drug is a term for the use of more than one drug or type of drug at the same time or one after another. However, I understand reviewer’s point. I rewrote the sentence: (Line 40-45) According to the National comprehensive cancer network (NCCN) guidelines for pancreatic adenocarcinoma [4], adjuvant gemcitabine monotherapy or adjuvant 5-fluorouracil (5-FU) monotherapy has been used for a long time for improvement after surgical resection. Moreover, adjuvant FOLFIRINOX (oxaliplatin, leucovorin, iri-notecan and 5 -FU) chemotherapy or a combination of gemcitabine and capecitabine should be considered for patients with good performance status [5].
REVIEWER: (Line 74, 75) Please clarify how can SREBP-1 be a predictor of poor prognosis in patients if its absence leads to inhibition of proliferation and induction of apoptosis?
Our response: Thank you for your comments. We revised the sentences as follows: (Line 76-80) In a previous paper, it was reported that SREBP-1 levels were significantly higher in pancreatic cancer than in normal tissue adjacent to the tumor through 60 pancreatic cancer patients. Moreover, high expression of SREBP1 indicated that associated with clinicopathological features such as a tumor differentiation, tumor-node-metastasis stage and lymphnode metastasis [20].
- Methods are missing catalogue numbers and producers of antibodies, kits and siRNAs, as well as PCR conditions, bioinformatic analysis description and western blot densitometry analysis.
Our response: We have added the catalogue numbers and producers of antibodies, kits and siRNAs. In addition, we have added more detailed descriptions for western blot densitometry analysis and bioinformatic analysis in Method section. However, PCR conditions information has been citated to follow the author's guidelines, concise articles.
REVIEWER:Thank you.
REVIEWER:PCR cycling conditions (temperature and cycle lenght) haven't been described in the paper that was cited. Please insert.
Our response: Thank you for your comments. We descibed for PCR cycling conditions in method section. (Line 183-186) Briefly, PCR amplifications were always performed in duplicate wells following the SYBRGreen manufacture Thermal Cycling Parameters: 95oC for 10 min on hold step, then 40 cycles were performed by two-step per cycle (95 oC for 15 sec on denaturing step and 60 oC for 60 sec on anneal step).
- Figures, as well as results would benefit from a more detailed description of methods. Results would benefit from small introduction or simple explanations why experiments were done.
Our response: The detailed description of methods and goal has been described in the text.
REVIEWER: Thank you.
REVIEWER: Figures would benefit from a more detailed description of methods, e.g. Fig 1B, D, E lacks crucial information about the method used.
Our response: Thank you for your comments. According to the guidelines of the pharmaceutical journal Pharmaceutics, "well-established methodologies can be briefly described and appropriately cited". As you previously commented, we wrote the catalog number and product information of the item in detail in the method section. Therefore, our description of immunoblotting and microarray analysis, which are well-known methods, is considered sufficient.
REVIEWER: Furthermore, part of title for Fig1: " and inhibits SREBP-1 transcriptional activity in PC cells. " is not correct, since transcriptional activity wasn't measured by any functional assay. Please soften the statement.
Our response: Thank you for your comments. We changed the “Treanscriptional activity” to “Expression” in Fig1 title.
- More specific points in results section: it is unclear how the authors chose the 5μM concentration for microarray experiments, what the table 1 presents, how did authors choose pathways to study further (metabolic), why the SREBP-1 was chosen for further analysis,
Our response: We have added more detailed descriptions in introduction (Line 62-75, page 2) and results section as reviewer pointed out (Line 283-284, page 7).
REVIEWER: Thank you.
REVIEWER: It is unclear how the authors chose the 5μM concentration for microarray experiments.
Our response: Thank you. As already you known about pharmaceutical biological experiments, drug concentrations that give 50-80% cell viability should be selected to perform PCR or Western blot to assess drug-regulated expression of proteins/RNAs. If the concentration is high, there will not be enough protein for the experiment. Because our drug induces apoptosis. Therefore, the reasons for choosing 5μM in our manuscript results section are explained as follows: (Line 269-273) “As shown in Fig. 1a, 10 µM of TA3 was reduced viability less than 50% compared to controls in all three cell lines. In particular, we found that BxPC-3 cells were induced cytotoxicity even at a low concentration of TA3 (1.25 µM). Therefore, we performed microarray analyses of BxPC-3 cells treated with TA3 (5 µM) for 24 h”.
REVIEWER: Table2 legend is lacking, it is unclear what it represents when compared to Fig1B.
Our response: Thank you. We reviesed the Table 2 legend as follows: The 22 genes most differentially expressed in TA3 treatment compared to untreated BxPC-3 cells were selected based on microarray results.
REVIEWER: Fig 1B) Maybe change" The number of gene transcripts in each category was listed." to "X-axis represent the number of gene transcripts in each category. Negative values represent downregulated and positive upregulated genes after TA3 treatment." or similar.
Our response: Thank you for your comments. As you pointed out, we have change the fig. 1b legend. X-axis represent the number of gene transcripts for 1.5-fold change regulates in each category. Negative values (blue bar) represent downregulated and positive values (red bar) represent upregulated genes after TA3 treatment.
REVIEWER: (Line 273) what is the meaning of " functional genes "? Please clarify or use different term.
Our response: Thank you for your comments. We have change the term and revised sentence as follows: (Line283-285) “As shown in Table 1, the 22 genes were classified by oncological function and over the 1.5-fold-change after TA3 treatment.”
REVIEWER: (Line 276) double!
Our response: Thank you! We deleted double sentence.
Additional: HMGCR results haven't been mentioned in the text.
Our response: Thank you! We added the mention about the HMGCR results as follows: (Line293-295) After TA3 treatment, the mRNA level of HMGCR was reduced by 1.2-fold in BxPC-3 cells, and 1.8-fold in PANC-1 cells, which is expressed by SREBP-2 and is involved in the cholesterol synthesis pathway such as HMGCS-1 [17].
REVIEWER: What does it mean: " In addition, we found that two genes were interacted with 20 genes, which was related with SREBP signaling pathway using GeneMANIA (Fig. s1)."? I guess this is the explanation to my question: Why the SREBP-1 was chosen for further analysis? But it is unclear. Please clarify or use different term.
Our response: Thank you for your comments. We have revised the sentence as follows: (Line: 295-301) To investigate INSIG-1 and HMGCS-1 interacting genes, we performed using GeneMANIA. Fig. s1 results are indicated that 20 genes related to INSIG-1 and HMGCS-1 including SREBF2, SREBF1, SCAP, HMGCR, HMGCS2, and INSIG2. We mentioned in introduction, SREBP bound by INSIG-1 remains an inactive precursor in the ER as the SCAP/SREBP complex. Therefore, we evaluated the expression of SREBPs to determine whether the increase in INSIG-1 caused by TA3 treatment is to regulate SREBPs.
REVIEWER: (Line 284) Please write here why you have decided to check nuclear and cytoplasmic fractions of SREBP-1. Insert more details with reference.
Our response: Thank you for your comments. In introduction section line (65-68) was descripted but, we more added in results section line (302-303) follow as: Mature form of SREBP-1 is migrated into the nucleus and activates promoters of lipogenic genes.
REVIEWER: (Line 289) missing reference: " SREBP-1 downstream targets "
Our response: Thank you. We added a reference 16.
- Some images lack densitometry and aication, like Fig s1 A and B, Fig 2C.
Our response: Unfortunately, we could not increase lack densitometry data from the specimens. Because In the case of a protein that appears in multiple forms on one membrane, it had to be stopped at an appropriate exposure in order to prevent the density of strong bands from becoming too saturated due to an increase in exposure.
REVIEWER: Thank you. Densitometry is done post-filming. Please analyse Fig s2 A and quantify S2B and Fig2C.
Our response: Thank you. We added the quantify Fig. s2A. Immunofluorescence results were performed to confirm the location of SREBP-1 expression in the nucleus, and the results were clearly indicate through the color of magenta combined with DAPI (Blue) and SREBP-1 (Red) even without quantitative graph. However, we agree with reviewer that images are subjective data. Therefore, we quantified the expression of SREBP-1 by separating the nucleus and cytoplasm using Western blotting for objective evaluation.
- Unclear about what upper part of fig2e means, unclear photo in lower part of 2F.
Our response: As reviewer pointed out, we revised the sentences as follows: To evaluate si-SREBP-1 whether induces knock down of SREBP-1 in BxPC-3 cells, we performed the immunoblotting and immunofluorescence. The results have shown that expression of SREBP-1 protein was silenced by si-SREBP-1 in BxPC-3 cells (Fig. 2e and f). (Line 320-330, page 9)
REVIEWER: It is still unclear what the upper part of fig2e means. Please explain what does the asterix mean in fig2e? and what the ns refers to. Also, comment this in the text.
Our response: Thank you. we understand what is reviewer point out. First, n.s. mean is not significance and there is no exist asteix it is p = 0.1. Thus, It might be make confused to reader, we decied to remove the n.s. and more descripted results follow as: Line(348-355) “The results shown that expression of SREBP-1 mature form and precuser form in nuclear was inhibited by si-SREBP-1 compared with scramble si-RNA as a control in BxPC-3 cells (Fig. 2e). Although the reduction of SREBP-1 was not significant due to high deviation obteined from independent triplicate experiments, in immunofluorescence results, nuclear localization of SREBP-1 was clearly reduced by si-SREBP-1 treatment in BxPC-3 cells (Fig. 2f). Thus, these results were represented that our si-SREBP-1 protocol was well silencing of SREBP-1 in BxPC-3 cells.”
Additional: Fig 2g: why wasn't pACC calculated relative to ACC in this case?
Our response: As mentioned in the background, SREBP-1 is a transcription factor expressing enzymes involved in fatty acid synthesis such as ACC and FASB. Therefore, we want to show a clear decrease in SREBP-1 expression with the levels of total ACC and phospho-ACC, respectively, relative to SREBP-1 being well silenced by siRNA. However, as pointed out, it was convincing to present only Total-ACC, so it was modified.
- Fig 2G. Missing explanation on why authors checked phospho protein?
Our response: We added the sentences for background of target proteins in the introduction section (Line 62-75, page 2) and briefly added them in the results section. (Line 330-332, page 9)
REVIEWER: Missing explanation on why authors checked phosphorilated form of the protein ACC? Please insert relevant reference.
Our response: Thanks for pointing it out. Phospho-ACC is known to be regulated by AMPK. However, in our results, the expression of AMPK by TA3 did not change significantly. As another mechanistic report, there is a report that AMPK activity suppresses the expression of ACC and FASN through downregulation of SREBP-1C [1]. In general, the expression of Phospho-ACC(s79) and ACC should be confirmed in a study of lipid suppression through AMPK activity [2, 3], but ACC expression or mRNA level should be indicated in a study of lipid suppression through SREBP-1 [4, 5]. Therefore, in agreement with the reviewers' comments, we presented only total-ACC expression to demonstrate that TA3 reduced the expression of ACC and FASN through SREBP-1 inhibition. (Supplementary PDF file for data blotting)
- Kohjima, M.; Higuchi, N.; Kato, M.; Kotoh, K.; Yoshimoto, T.; Fujino, T.; Yada, M.; Yada, R.; Harada, N.; Enjoji, M.; Takayanagi, R.; Nakamuta, M., SREBP-1c, regulated by the insulin and AMPK signaling pathways, plays a role in nonalcoholic fatty liver disease. Int J Mol Med 2008, 21, (4), 507-11.
- Fan, K.; Lin, L.; Ai, Q.; Wan, J.; Dai, J.; Liu, G.; Tang, L.; Yang, Y.; Ge, P.; Jiang, R.; Zhang, L., Lipopolysaccharide-Induced Dephosphorylation of AMPK-Activated Protein Kinase Potentiates Inflammatory Injury via Repression of ULK1-Dependent Autophagy. Front Immunol 2018, 9, 1464.
- Lee, K.; Lee, Y. J.; Kim, K. J.; Chei, S.; Jin, H.; Oh, H. J.; Lee, B. Y., Gomisin N from Schisandra chinensis Ameliorates Lipid Accumulation and Induces a Brown Fat-Like Phenotype through AMP-Activated Protein Kinase in 3T3-L1 Adipocytes. Int J Mol Sci 2020, 21, (6).
- Kim, Y. M.; Shin, H. T.; Seo, Y. H.; Byun, H. O.; Yoon, S. H.; Lee, I. K.; Hyun, D. H.; Chung, H. Y.; Yoon, G., Sterol regulatory element-binding protein (SREBP)-1-mediated lipogenesis is involved in cell senescence. The Journal of biological chemistry 2010, 285, (38), 29069-77.
- Meng, H.; Shen, M.; Li, J.; Zhang, R.; Li, X.; Zhao, L.; Huang, G.; Liu, J., Novel SREBP1 inhibitor cinobufotalin suppresses proliferation of hepatocellular carcinoma by targeting lipogenesis. Eur J Pharmacol 2021, 906, 174280.
REVIEWER: (Line 330) please change inhibited to "lowered the expression" or similar.
Our response: Thank you. We were changed it.
REVIEWER: (Line 331) Please change " increased the expression " to increased"
Our response: Thank you. We were changed it.
- What do the arrows in 2C IF images point to?
Our response: As on the manuscript, we explained arrows mean in figure legend as follows: “Arrows denote nuclear translocation”. In addition, we has been described it in results as follows: “We also found that treatment of BxPC-3 cells with TA3 suppressed the nuclear translocation of transgenic anti-SREBP-1-REP and was reduced its level in the cytosol (Fig. 2c).” (Line 321-323, page 9)
REVIEWER: Thank you. I see now. Please explain what is "transgenic anti-SREBP-1-REP"?
Our response: Thank you for your comments. we revised “trasgenic anti-SREBP-1-RFP” to “translocation of SREBP-1 was suppresed translocation into the nulceus by TA3 treatment and showed weak exposure in the cytoplasm compared to the control.” (Line 341-343)
- Figure 3A and B show different percentages which differ from text.
Our response: Thank you for your comments. Our cell cycle graph has been represented the independent triplicate experiments. Therefore, cell cycle distribution percentage on Fig 3b looks like a different with Fig 3a by average value of triplicate experiments.
REVIEWER: Thank you. This information should be added in the description of 3a.
Our response: Thank you for your comments. we added the sentences as follows: (Line 377-380) “In the cell cycle assay results, our cell cycle graph has been represented the independent triplicate experiments. Therefore, the results of cell cycle distribution in Fig. 3a showed one of three independent experiments. Then, we graphed the mean of three independent experiments.”
- Invalid statement: Our results showed that TA3 treatment blocked Akt/Gsk3b signaling-dependent decrease in SREBP-1 maturation, whereas no changes were observed in the phosphorylation of mTOR (Fig. 4a and b). These results didn’t investigate SREBP-1 maturation.
Our response: As reviewer pointed out, We have revised the manuscript due to misleading expressions in the text.(Line 390-392, page 13)
REVIEWER: Thank you. Added part of the sentence: "SREBP-1 inhibition was induced" is unclear and added " GSK3b were inhibited " is incorrect, since data show increase of phosphoGSK3b. Please clarify or use different term.
Our response: Thank you for your comments. GSK3β is phosphorylated into an inactive state at Ser9. We revised the sentences as follows: (Line 417-425)“First, we evaluated the effect on phosphorylation of Akt and Gsk3β after 12 h of TA3 treatment. Phosphorylation of Akt at Ser473 was slightly increased at the 2.5 μM of TA3, but was markdely inhibited by TA treatment at the 5 μM compare with untreated control. Increased phosphorylation of Gsk3β at Ser9 was increased by TA3 treatment at the both concentration compare with untreated control. Gsk3 has been known to be an enzyme that phosphorylates and inactivates glycogen synthase [29]. Therefore, our results showed that TA3 treatment significantly induced GSK3β inactivation.”
- Quantification of Fig 5C is unconvincing concerning volumes of GEM vs. vehicle treated. It seems as though the volumes of GEM-treated tumors are bigger then vehicle.
Our response: When viewed as a picture in Fig 5c, the GEM tumor looks not much different from the vehicle because, GEM tumor shape is flat. However, when the size is measured, the difference appears as in the result of Fig 5d.
REVIEWER: Thank you. This should be written in the text describing results in Line 433 after " compared to the vehicle group (Fig. 5c and d)."
Our response: Thank you. We added the text in Line 466-468 after “compared to the vehicle group(Fig. 5c and d).
- Interpretations for fig 5b are unconvincing. Quantification should be done. Incorrect conclusion: “these results demonstrated that the inhibition of srebp-1 in the TA3-treated subcutaneous BxPC-1 tumor reduced fatty acid synthesis” conclusions for subcutaneous BxPC-3 tumor cannot be made based on cell culture data. Furthermore, since 5uM TA3 causes cell death in BxPC3 cells, one should be careful about interpreting lower lipid content (Fig 1A)
Our response: We have added the sentences for “preparation of sample and fatty acid standards” in method section. We extracted the metabolites from all mouse tumors with equal weight (5 mg/ml) and quantified them. (Line 224-233, page 5-6)
REVIEWER: Thank you.
REVIEWER: Incorrect conclusion here, please revise: “these results demonstrated that the inhibition of srebp-1 in the TA3-treated subcutaneous BxPC-1 tumor reduced fatty acid synthesis” Conclusions about fatty acids for subcutaneous BxPC-3 tumor cannot be made based on cell culture data! furthermore, synthesis of fatty acids haven't been checked. ORO staining only can checks abundance or content, but not synthesis!
Our response: Thank you for your comments. As your point out, we chaged the sentence as follows: (Line 513-515) “These results implicate that the inhibition of SREBP-1 in the TA3-treated subcutaneous BxPC-3 tumor reduced lipid metabolism.”
REVIEWER: Fig 3 d doesn't have a legend.
Our response: We have a Fig. 3d legend line 409-410, “(d) The effects of TA3 on the p27 and p21 expression levels, related to cell cycle arrest were detected by immunoblotting.” We've highlighted it for reviewer.
REVIEWER:Line 481: Lipogenesis hasn't been checked, please revise this statement to something softer (probable, possible or suggesting inhibition): "through inhibition of lipogenesis."
Our response: Thank you. As you well know, our results like a suppression of SREBPs, FASN, and ACC expression, also reduction of saturated fatty acids level were indirect proofs that the fatty acid synthesis is deregulated. However, as you point out, we understood there might be some misunderstandings in the expression thus we revised the sentence as follows: (Line 516-519) “also the results of reduced palmitate and stearate level in subcutaneous BxPC-3 tumor supported the SREBP-1 inhibitory effect of TA3 in cell experiments. Thus, we suggested that TA3 has an anti-lipogenic effect through inhibition of SREBP-1 and enzymes involved with lipid metabolism.”
REVIEWER:Fig 6 title is missleading on "synthesis" since synthesis hasn't been checked: " TA3 inhibitory effects against fatty acid synthesis in pancreatic tumor tissues. " Please change to "TA3 lowers fatty acid content in pancreatic tumor " or similar.
Our response: We changed the sentence as follows: Figure 6. TA3 inhibits SREBP-1 and FASN expression, which are involved in the lipogenesis pathway, and lowers fatty acids levels in pancreatic tumor tissues.
- Unconvincing statement: “which suggests that the cell growth inhibitory effect of TA3 is dependent on SREBP-1-related lipogenesis” Use of insulin in experiments should be further clarified.
Our response: There is some reason in what the reviewer pointed out. Therefore, we added more detailed description in main text for understand our hypothesis and results.
REVIEWER: Line 531: “related lipogenesis” The results that came before this sentence don't discuss about lipids so this conclusion is incorrect. Also, lipogenesis hasn't been tested.
Our responsed: Thank you. We revised the sentences as follows: (Line 571-574) The level of cleaved PARP was increased following silencing of SREBP-1 in BxPC-3 cells, and this result was the same as that observed in TA3-mediated apoptosis, which suggests that the cell growth inhibitory effect of TA3 is dependent on expression of SREBP-1.
- Title is misleading: transcriptional activity of SREBP-1 as well as the fatty acid biosynthesis haven’t been checked per se.
Our response: There is some reason in what the reviewer pointed out. However, we confirmed a decrease in fatty acid synthase activity such as FASN and ACC through TA3-induced inhibition of SREBP-1 in BxPC-3 cells. Moreover, we suggested reduced levels of saturated fatty acids such as palmitate and malate in tumor tissue due to TA3 treatment.
REVIEWER: The paper has given indirect proofs that the fatty acid synthesis is deregulated and that the SREBP-1's transcriptional activity has been hindered. But the paper doesn't provide direct functional experiments which checked
-transcriptional activity of SREBP-1 nor
-fatty acid biosynthesis
and thus it would be of benefit that the title softens a bit.
Our response: Thank you. We chaged the title “Timosaponin A3 Inhibits Palmitate and Stearate through Suppression of SREBP-1 in Pancreatic Cancer”
Related to this:
-Please, see if you may remove transcriptional activity from figure legend Fig1 to something more suitable like: " TA3 regulates lipid metabolism, lowers the nuclear localization of SREBP-1 and the expression of its target genes in PC cells. "
-Please, see if you may remove transcriptional activity from figure legend Fig1 to something more suitable like: " TA3 regulates lipid metabolism and inhibits SREBP-1 and its target genes in PC cells. "
Our response: Thank you. We chaged the figure 1 legend as follows: TA3 regulates lipid metabolism and inhibits SREBP-1 expression in PC cells.
-Please, see if you may remove transcriptional activity from figure legend Fig2 to something more suitable like: " TA3 reduces mature SREBP-1 in BxPC-3 cells "
Our response: Thank you. We chaged the figure 2 legend as follows: TA3 reduces the nuclear localization of mature SREBP-1, which inhibits lipogenic enzymes and cell growth in BxPC-3 cells.
-Line 477: Please change if possible : "These results demonstrated that the inhibition of SREBP-1 in the TA3- " to: "These results implicate" or "These results suggest" or similar..
Our response: Thank you. As your comments, we chaged the sentence as follows: (Line 513-515) “These results implicate that the inhibition of SREBP-1 in the TA3-treated subcutaneous BxPC-3 tumor reduced lipid metabolism.”
-figure 6. legend: Please omit (delete) "synthesis " if possible: " TA3 inhibitory effects against fatty acid synthesis in pancreatic tumor tissues.
Our response: Thank you. We chaged the figure 6. legend as follows: TA3 inhibits SREBP-1 and FASN expression, which are involved in lipogenesis pathway, and lowers fatty acid content in pancreatic tumor tissues.
Additional remarks:
-Line 521: Please correct the reference: " Maria T. and Johan"
Our response: Thank you. We corrected the reference as your comments.
-Line 345: correct this: " expression of SREBP-1 nuclear translocation " to "expression of nuclear SREBP-1" or similar.
Our response: Thank you. We revised the sentence as follows: “to evaluate the expression of SREBP-1 in nucleus.”
-Fig 5.A text "as described in Materials and Methods." is inappropriate. figure has to be "stand alone" and all the important info has to be written, without references to other parts of the paper.
Our response: Thank you. We descibed in figure legend as follows: Briefly, 1 x 107 cells of BxPC3 were mixed with Matrigel at a ratio 1:1 (v/v) before subcutaneously injection and randomly assigned to six mice of each group. 100 uL of PBS and each drugs (5mg/kg of TA3 and GEM) were intraperitoneally injected to mice of each group three times a week for 20 days with a measure of tumor volume and euthanized five days after the end of treatment.
-Line 507: Please check if "overexpression" here is correct or add details or refenrence to figure to make it clearer.
Our response: Thank you for your comments. We added the reference and revised the Line 550 as follows: high expression of SREBPs in PC [20].
-Line 514: Please check if "on the nucleic" is correct.
Our response: Thank you. We chaged it “on the nucleus in BxPC-3 cells”
-Line 516, 517: "reduction..is induced" Please change to a more suitable wording.
Our response: Thank you. We revised sentence as follows: (Line 557-558) “TA3 suppressed mature SREBP-1 and SREBP-2 expression on the nucleus in BxPC-3 cells and reduced level of lipid accumulation in 3T3-L1 and BxPC-3 cells.”
-Line 553: " inhibitory effect of Akt and GSK3β activity by TA3" is incorrect for GSK3B, since your data show that TA3 activates GSK3B! Same comment for Line 583:" TA3 reduced Akt/GSK3β signaling "
Our response: Thank you. We revised sentence as follows: (Line 596-597) whereas the suppression of Akt phosphorylation and induction of GSK3β inactivation due to increased Ser9 phosphorylation after TA3 treatment was blocked by insulin.
(Line 626-631) Mechanistically, TA3 redcued Akt-Ser473 phosphorylation and increased GSK3β-Ser9 phosphorylation and enhanced GSK3β inactive, but the TA3 inhibitory effect on SREBP-1 was independent of the regulation of Akt and GSK3β phosphorylation.
-Line 560: missing reference In recent years, interest in natural extracts as anticancer agents has been of considerable interest as they offer the potential to discover many active compounds and significantly reduce chemotherapy-related toxicities by replacing them with naturally derived ingredients.
Our response: Thank you. We added the reference 46.
-Line 416 and 388: I wonder why was the conclusion made that the inhibition of SREBP-1 TA3 appears independent of the Akt/Gsk3β mechanism? Please add explanation.
Our response: We explaned the why inhibition of SREBP-1 TA3 appears independent of Akt/Gsk3β pathway in discussion section (line 597-599). Nonetheless, as you comments we added explanation in summary (conclusion line 629-635) paragraph as follows: Althought the suppression of Akt phosphorylation and induction of GSK3β-Ser9 inactivation by TA3 was blocked by pretreatment with insulin. The inhibitory effect of mSREBP-1, FASN, and ACC expression by TA3 was maintained even on pretreatment with insulin as much as when TA3 was treated alone. Thus, our results established that the TA3 inhibitory effect on SREBP-1 was independent of the regulation of Akt and GSK3β phosphorylation. Taken together, We propose that TA3 might be a direct inhibitor of SREBP-1, capable of inducing inhibition of fatty acid synthesis and tumor growth.
-Line 565: change " enzymes " and " factors " to singular if correct.
Our response: Thank you. We corrected to singular.

Reviewer 3 Report
- Line 37: the numbers of new cancer cases and deaths are each year estimated. There is a very new publication for 2022, the five-year survival rate is now at 11% (PMID: 35020204).
Our response: Thank you for your comments. We follow the reviewer suggestion. (Line 37, page 1)
Please check your citation list again. If the information comes from the publication, please cite it in addition.
- Line 42: Are GEM and 5-FU still standard options? How about FOLFIRINOX? Please include recent publication or guideline.
Our response: As reviewers have noted, FOLFIRINOX is also being used as a treatment. In this study, GEM was mentioned using GEM as a standard material, and GEM was selected as a standard material because it is a relatively inexpensive drug and is generally used in many papers. We added your comments in introduction section. (Line 43, page 1)
The newly added sentences are not clear. I agree that GEM therapy is considered as an option. But, what do the recent guidelines say? Which therapeutic options are available?
- Line 70 (Methods): please include methods for western blotting.
Our response: The method for western blotting commented by the reviewer is written in section 2.7 cell lysis and immunoblotting in the original manuscript. (Line 148-161, page 4)
The authors answered my question.
- Line 237 (Figure 1d): “The expression level of the mature form of SREBP-1 was decreased in all three cell lines following TA3 treatment” The data is not convincing. Please perform the experiment again and re-analyze the data. To be sure that cellular fractionation worked. Please include nuclear and cytosol markers in your western blotting as well. Also, please label the size of your bands (kDa)
Our response: We agree with the reviewer’s opinion. Unfortunately, the author who took the lead in this experiment has been promoted and is in a different place, so it is difficult to proceed with the experiment again. The goal of this results is to find the connection between SREBP-1 and TA3 in pancreatic cancer cells and which cell is most regulated by TA3. We suggest that SREBP-1 expression was decreased 0.9-fold (AsPC-1), 0.6-fold (BxPC3), and 0.8-fold (PANC-1) by TA3 in figure 1d. We have revised sentences in the revised manuscript as reviewer pointed out. (Line 286-288, page 7)
I think this is a key experiment for your project. Just simply changing the sentence is not the option. Please address the concern experimentally.
- Line 240 (Figure 1e): did author analyze ACC1 or ACC2? Which phosphorylation site did the author analyze (presumably s79, then ACC1?)? AMPK phosphorylates and directly inhibits ACCs. Somewhat I do not understand why the authors try to show less FASN expression and less P-ACC.
Our response: We have added the target proteins information in introduction section for understand. (Line 66-72, page 2) In addition, we have revised the sentences in results as reviewer pointed out. (Line 288-291, page 7)
“SREBP-1 is regulated” would cause misinterpretation that SREBP-1 is a downstream target. Please revise the introduction part accordingly. The authors added some information about SREBP-1, SREBP-2-dependent regulation. But, the authors did not answer my question. This is an important point.
- Line 261 (Figure 2a): ACC phosphorylation results in its inhibition leading to suppression of lipogenesis. Here I do not understand why the authors try to show FASN downregulation and less ACC phosphorylation.
Our response: We have added the information for target proteins as reviewer pointed out. (Line 66-72, page 2) In addition, we have changed the sentences “we incubated BxPC-3 cells with TA3 (2.5 and 5 μM) for 12 h and performed western blotting to confirm the expression levels of FASN and phospho-ACC, two regulators of fatty acid synthase” to “we incubated BxPC-3 cells with TA3 (2.5 and 5 μM) for 12 h and performed western blotting to confirm whether the expression levels of FASN and phospho-ACC, two regulators of fatty acid synthase are regulated by TA3 dose-dependent manner” throughout the text. (Line 307-309, page 9)
Same to the point 5, the authors did not answer my question.
- Line 275 (Figure 2d): The authors labeled pSREBP-1 was from the nucleus, in figure 1d the authors labeled pSREBP-1 was form cytosol.
Our response: The SREBP-1 forms are presented as precursor form (125kDa) in the cytosol and the mature form (68kDa) in the nucleus by western blots. Although the precursor form of SREBP-1 (125kDa) is thought to be localized to the endoplasmic reticulum membrane, previously studied (Wang et al. Cell 1994;77(1):53-62, Wu et al. Am J Physiol 1999;277(6):e1087-94) noted that precursor form of SREBP-1 form could be shown in the salt nuclear extract in HeLa cells like our results. Also, we can find similar results on the company of anti-body products.
If pSREBP-1 can be from the nucleus, then it is very important that the author experimentally evaluate whether the authors really have either nuclear or cytoplasmic fractions (also figure 1d).
- Line 280 (Figure 2g): same to the points 6 and 7. The authors showed less ACC phosphorylation after siRNA.
Our response: According to the reviewer's comments, we have added information about the target protein to the introduction in the same way as in answers 6 and 7. (Line 62-75, page 2) Also, we have revised the sentences in the results section (Line330-334, page 9) as reviewer pointed out.
The newly added introduction part did not answer the question.
- Line 280 (Figure 2h): while western blot data for p27 and p21 are convincing, blotting data for PARP is not convincing.
Our response: We have revised the sentences and replaced blotting data in the revised manuscript as reviewer pointed out as follows: Silencing of SREBP-1 significantly inhibited lipid metabolism-related enzymes, and in-creased the expression of cleaved PARP (1.4-fold) and p21 (1.9-fold) compared to control (Fig. 2g and h). Although no significant results were obtained due to high deviation, the expression of p27 was greatly increased 2.9-fold by si-SREBP-1. (Line 330-334, page 9)
The authors addressed the concern.
- Line 332 (Figure 4a): Blotting data for p-Akt is convincing, but the other blots are not convincing.
Our response: We have replaced blotting data as reviewer pointed out. In addition, we have been mentioned “whereas no changes were observed in the phosphorylation of mTOR (Fig. 4a and b).” in results section. And we have revised the sentences as reviewer pointed out as follow: “phosphorylation of AMPK was slightly increased following TA3 (5 μM) treatment.” (Line 390-394, page 13)
The authors did not replace any blotting data.
- Line 341 (Figure 4d): Same to the point 7, pSREBP-1 in the nucleus?
Our response: Same as answer point 7, the SREBP-1 forms are presented as precursor form (125 kDa) in the cytosol and the mature form (68 kDa) in the nucleus by western blots. Although the precursor form of SREBP-1 (125 kDa) is thought to be localized to the endoplasmic reticulum membrane, previously studied (Wang et al. Cell 1994;77(1):53-62, Wu et al. Am J Physiol 1999;277(6):e1087-94) noted that precursor form of SREBP-1 form could be shown in the salt nuclear extract in HeLa cells like our results. Also, we can find similar results on the company of anti-body products.
Yes, therefore the authors should perform experiments to clarify with nuclear and cytosol markers to show whether the authors have pure nuclear or cytosol fraction.
- Line 347 (Figure 4e): Same to the point 6, less FASN and less p-ACC?
Our response: We have added the target proteins information in the introduction section for understand. (Line 62-75, page 2)
Same for previous points, the information is not addressing my question.
- Line 363 (Figure 5a): the schema shows that “BxPC3 injection at D-7, randomization D-0, treatment stop D-20, and at D-25 sacrifice”. In the method section, the authors described that “euthanized 1 week after the end of treatment”. What exactly did the author perform the experiment? If the method part is correct, please change the scheme Figure 5a. Further, how the authors know “the tumors reached a diameter of 0.5 cm"? Does it take exactly 7 day as in Figure 5a? How did the authors exactly know tumor volume between days 2 to 18?
Our response: Thank you for your comments, we have revised 5 days in the method section. (Line 205, page 5) As on the original manuscript, we explained the how tumor volume was measured in the method section (Line 204-207, page 5).
- Please add information that the authors randomized 7 days after injection in addition.
- Line 386 (Figure 5f): Please include statistical analysis in the figure. Is it IHC for cleaved caspase 3?
Our response: Sorry for any inconvenience. We used cleaved caspase 3 in IHC assay as we mentioned in the materials and method section. We have revised “caspase 3” to “cleaved caspase 3” in the figure 5f and figure legend. IHC staining showed a clear difference in protein expression level between vehicle and TA3 slides even without statistical analysis.
Still please add statistical analysis.
Author Response
Hyeung Jin Jang, Ph.D.
Professor
Department of Biochemistry,
College of Korean Medicine
Kyung Hee University,
26, KungHeedae-ro, Dongdaemun-gu,
Seoul, 02447, Republic of Korea
E-mail: [email protected]
Apr 18th, 2022
Manuscript ID: Pharmaceutics-1564084
Revised tittle:
Timosaponin A3 Inhibits Palmitate and Stearate through Suppression of SREBP-1 in Pancreatic cancer
Dear Editors,
Thanks to the reviewers and Pharmaceutics for taking the time to review the article. Some additional experiments, corrections, and clarifications were made to the manuscript after receiving comments from the judges.
In the revised manuscript, each reviewer's comments were carefully answered one by one, just like the authors' responses. We hope that the revised manuscript better meets the journal's requirements for publication. We would like to thank the editors and reviewers of Pharmarceutics once again for a constructive review of our paper.
Sincerely yours,
Hyeung-Jin Jang
Comments and Suggestions for Authors:
Reviewer #3
- Line 37: the numbers of new cancer cases and deaths are each year estimated. There is a very new publication for 2022, the five-year survival rate is now at 11% (PMID: 35020204).
Our response: Thank you for your comments. We follow the reviewer suggestion. (Line 37, page 1)
REVIEWER: Please check your citation list again. If the information comes from the publication, please cite it in addition.
Our response: Sorry, we checked and the references didn't put into it, so we put it back in.
- Line 42: Are GEM and 5-FU still standard options? How about FOLFIRINOX? Please include recent publication or guideline.
Our response: As reviewers have noted, FOLFIRINOX is also being used as a treatment. In this study, GEM was mentioned using GEM as a standard material, and GEM was selected as a standard material because it is a relatively inexpensive drug and is generally used in many papers. We added your comments in introduction section. (Line 43, page 1)
REVIEWER: The newly added sentences are not clear. I agree that GEM therapy is considered as an option. But, what do the recent guidelines say? Which therapeutic options are available?
Our response: Thank you. According to the NIH treatment and National comprehensive cancer network of pancreatic cancer, historically, multiple randomized trials have established that adjuvant gemcitabine monotherapy or adjuvant 5-FU monotherapy improve OS for 6 months after surgical resection compared with surgery alone. More recent studies have looked at newer combination regimens that might further improve outcomes after surgical resection.
For patients with good performance status, adjuvant FOLFIRINOX (oxaliplatin, leucovorin, irinotecan, and 5-FU) chemotherapy or the combination of gemcitabine and capecitabine should be considered. However, for older patients or patients with marginal performance status, adjuvant gemcitabine or 5-FU monotherapy can be considered. In Asia, S-1 (tegafur, gimeracil, and oteracil potassium) is an appropriate alternative to gemcitabine-based therapies.
Thus, we added this information in introduction section (Line 40-45).
- Line 237 (Figure 1d): “The expression level of the mature form of SREBP-1 was decreased in all three cell lines following TA3 treatment” The data is not convincing. Please perform the experiment again and re-analyze the data. To be sure that cellular fractionation worked. Please include nuclear and cytosol markers in your western blotting as well. Also, please label the size of your bands (kDa)
Our response: We agree with the reviewer’s opinion. Unfortunately, the author who took the lead in this experiment has been promoted and is in a different place, so it is difficult to proceed with the experiment again. The goal of this results is to find the connection between SREBP-1 and TA3 in pancreatic cancer cells and which cell is most regulated by TA3. We suggest that SREBP-1 expression was decreased 0.9-fold (AsPC-1), 0.6-fold (BxPC3), and 0.8-fold (PANC-1) by TA3 in figure 1d. We have revised sentences in the revised manuscript as reviewer pointed out. (Line 286-288, page 7)
REVIEWER: I think this is a key experiment for your project. Just simply changing the sentence is not the option. Please address the concern experimentally.
- Line 240 (Figure 1e): did author analyze ACC1 or ACC2? Which phosphorylation site did the author analyze (presumably s79, then ACC1?)? AMPK phosphorylates and directly inhibits ACCs. Somewhat I do not understand why the authors try to show less FASN expression and less P-ACC.
Our response: We have added the target proteins information in introduction section for understand. (Line 66-72, page 2) In addition, we have revised the sentences in results as reviewer pointed out. (Line 288-291, page 7)
REVIEWER: “SREBP-1 is regulated” would cause misinterpretation that SREBP-1 is a downstream target. Please revise the introduction part accordingly. The authors added some information about SREBP-1, SREBP-2-dependent regulation. But, the authors did not answer my question. This is an important point.
We revised the sentence “SREBP-1 regulates enzymes involved in fatty acid synthesis such as acetyl-CoA carboxylase (ACC), and fatty acid synthase (FASN).”
Thanks for pointing it out. As noted by reviewers, Phospho-ACC is known to be regulated by AMPK. However, in our results, the expression of AMPK by TA3 did not change significantly. As another mechanistic report, there is a report that AMPK activity suppresses the expression of ACC and FASN through downregulation of SREBP-1C [1]. In general, the expression of Phospho-ACC(s79) and ACC should be confirmed in a study of lipid suppression through AMPK activity [2, 3], but ACC expression or mRNA level should be indicated in a study of lipid suppression through SREBP-1 [4, 5]. Therefore, in agreement with the reviewers' comments, we presented only total-ACC expression to demonstrate that TA3 reduced the expression of ACC and FASN through SREBP-1 inhibition. (Figure 1e, and Supplementary PDF file for data blotting)
- Kohjima, M.; Higuchi, N.; Kato, M.; Kotoh, K.; Yoshimoto, T.; Fujino, T.; Yada, M.; Yada, R.; Harada, N.; Enjoji, M.; Takayanagi, R.; Nakamuta, M., SREBP-1c, regulated by the insulin and AMPK signaling pathways, plays a role in nonalcoholic fatty liver disease. Int J Mol Med 2008, 21, (4), 507-11.
- Fan, K.; Lin, L.; Ai, Q.; Wan, J.; Dai, J.; Liu, G.; Tang, L.; Yang, Y.; Ge, P.; Jiang, R.; Zhang, L., Lipopolysaccharide-Induced Dephosphorylation of AMPK-Activated Protein Kinase Potentiates Inflammatory Injury via Repression of ULK1-Dependent Autophagy. Front Immunol 2018, 9, 1464.
- Lee, K.; Lee, Y. J.; Kim, K. J.; Chei, S.; Jin, H.; Oh, H. J.; Lee, B. Y., Gomisin N from Schisandra chinensis Ameliorates Lipid Accumulation and Induces a Brown Fat-Like Phenotype through AMP-Activated Protein Kinase in 3T3-L1 Adipocytes. Int J Mol Sci 2020, 21, (6).
- Kim, Y. M.; Shin, H. T.; Seo, Y. H.; Byun, H. O.; Yoon, S. H.; Lee, I. K.; Hyun, D. H.; Chung, H. Y.; Yoon, G., Sterol regulatory element-binding protein (SREBP)-1-mediated lipogenesis is involved in cell senescence. The Journal of biological chemistry 2010, 285, (38), 29069-77.
- Meng, H.; Shen, M.; Li, J.; Zhang, R.; Li, X.; Zhao, L.; Huang, G.; Liu, J., Novel SREBP1 inhibitor cinobufotalin suppresses proliferation of hepatocellular carcinoma by targeting lipogenesis. Eur J Pharmacol 2021, 906, 174280.
- Line 261 (Figure 2a): ACC phosphorylation results in its inhibition leading to suppression of lipogenesis. Here I do not understand why the authors try to show FASN downregulation and less ACC phosphorylation.
Our response: We have added the information for target proteins as reviewer pointed out. (Line 66-72, page 2) In addition, we have changed the sentences “we incubated BxPC-3 cells with TA3 (2.5 and 5 μM) for 12 h and performed western blotting to confirm the expression levels of FASN and phospho-ACC, two regulators of fatty acid synthase” to “we incubated BxPC-3 cells with TA3 (2.5 and 5 μM) for 12 h and performed western blotting to confirm whether the expression levels of FASN and phospho-ACC, two regulators of fatty acid synthase are regulated by TA3 dose-dependent manner” throughout the text. (Line 307-309, page 9)
REVIEWER: Same to the point 5, the authors did not answer my question.
Our response: Thanks for pointing it out. Our answer to Figure 2a is the same as the point 5 response. (Supplementary PDF file for data blotting)
- Line 275 (Figure 2d): The authors labeled pSREBP-1 was from the nucleus, in figure 1d the authors labeled pSREBP-1 was form cytosol.
Our response: The SREBP-1 forms are presented as precursor form (125kDa) in the cytosol and the mature form (68kDa) in the nucleus by western blots. Although the precursor form of SREBP-1 (125kDa) is thought to be localized to the endoplasmic reticulum membrane, previously studied (Wang et al. Cell 1994;77(1):53-62, Wu et al. Am J Physiol 1999;277(6):e1087-94) noted that precursor form of SREBP-1 form could be shown in the salt nuclear extract in HeLa cells like our results. Also, we can find similar results on the company of anti-body products.
REVIEWER: If pSREBP-1 can be from the nucleus, then it is very important that the author experimentally evaluate whether the authors really have either nuclear or cytoplasmic fractions (also figure 1d).
Our response: Thank you for your comments. As you pointed out, we performed experiments to clarify with nuclear marker and presented Histone H3 in the nuclear fraction.
- Line 280 (Figure 2g): same to the points 6 and 7. The authors showed less ACC phosphorylation after siRNA.
Our response: According to the reviewer's comments, we have added information about the target protein to the introduction in the same way as in answers 6 and 7. (Line 62-75, page 2) Also, we have revised the sentences in the results section (Line330-334, page 9) as reviewer pointed out.
REVIEWER: The newly added introduction part did not answer the question.
Our response: Thanks for pointing it out. Our answer for Figure 2g is the same as the point 5 response. (Supplementary PDF file for data blotting)
- Line 332 (Figure 4a): Blotting data for p-Akt is convincing, but the other blots are not convincing.
Our response: We have replaced blotting data as reviewer pointed out. In addition, we have been mentioned “whereas no changes were observed in the phosphorylation of mTOR (Fig. 4a and b).” in results section. And we have revised the sentences as reviewer pointed out as follow: “phosphorylation of AMPK was slightly increased following TA3 (5 μM) treatment.” (Line 390-394, page 13)
REVIEWER: The authors did not replace any blotting data.
Our response: Thank you for your comments. In our results, the expression of AMPK and mTOR by TA3 did not much change significantly. Thus, blotting data for AMPK and mTOR are convincing. Although the blotting data for GSK3β was previously changed, the reviewer pointed it out again, so we replaced the blotting data of AMPK and GSK3β.
- Line 341 (Figure 4d): Same to the point 7, pSREBP-1 in the nucleus?
Our response: Same as answer point 7, the SREBP-1 forms are presented as precursor form (125 kDa) in the cytosol and the mature form (68 kDa) in the nucleus by western blots. Although the precursor form of SREBP-1 (125 kDa) is thought to be localized to the endoplasmic reticulum membrane, previously studied (Wang et al. Cell 1994;77(1):53-62, Wu et al. Am J Physiol 1999;277(6):e1087-94) noted that precursor form of SREBP-1 form could be shown in the salt nuclear extract in HeLa cells like our results. Also, we can find similar results on the company of anti-body products.
REVIEWER: Yes, therefore the authors should perform experiments to clarify with nuclear and cytosol markers to show whether the authors have pure nuclear or cytosol fraction.
Our response: Thank you. As you pointed out, we conducted experiments to clarify with nuclear marker and presented Histone H3 in the nuclear fraction.
- Line 347 (Figure 4e): Same to the point 6, less FASN and less p-ACC?
Our response: We have added the target proteins information in the introduction section for understand. (Line 62-75, page 2)
REVIEWER: Same for previous points, the information is not addressing my question.
Our response: Thanks for pointing it out. Our answer to Figure 4e is the same as the point 5 response. (Supplementary PDF file for data blotting)
- Line 363 (Figure 5a): the schema shows that “BxPC3 injection at D-7, randomization D-0, treatment stop D-20, and at D-25 sacrifice”. In the method section, the authors described that “euthanized 1 week after the end of treatment”. What exactly did the author perform the experiment? If the method part is correct, please change the scheme Figure 5a. Further, how the authors know “the tumors reached a diameter of 0.5 cm"? Does it take exactly 7 day as in Figure 5a? How did the authors exactly know tumor volume between days 2 to 18?
Our response: Thank you for your comments, we have revised 5 days in the method section. (Line 205, page 5) As on the original manuscript, we explained the how tumor volume was measured in the method section (Line 204-207, page 5).
REVIEWER: Please add information that the authors randomized 7 days after injection in addition.
Our response: Thank you for your comments. We added the information as follows: “After 7 days, we randomized six mice to each group …”
- Line 386 (Figure 5f): Please include statistical analysis in the figure. Is it IHC for cleaved caspase 3?
Our response: Sorry for any inconvenience. We used cleaved caspase 3 in IHC assay as we mentioned in the materials and method section. We have revised “caspase 3” to “cleaved caspase 3” in the figure 5f and figure legend. IHC staining showed a clear difference in protein expression level between vehicle and TA3 slides even without statistical analysis.
REVIEWER: Still please add statistical analysis.
Our response: Thank you. We added the statistical analysis of IHC assay including descriptions of methods in the Materials and Methods section.

Round 3
Reviewer 3 Report
The authors Kim et al. have submitted second revised version of the manuscript. While the authors successfully revised introductory section, they could not address the concerns raised by this reviewer in a way of additional experiments (especially about clarification of the role of phosphorylated ACC in the manuscript).
Since the authors keep the data in Figure 1d, “TA3… inhibits SREBP-1 expression” should be changed to “tends to inhibit” or similar, because the data is not significant.
Molecular weight labeling is still missing for figure 4a, 4c, and 4d.
Line 455: “… independent of Ask…” “Akt” instead of “Ask”?
Author Response
Apr. 20th, 2022
Manuscript ID: Pharmaceutics-1564084
Revised title: Timosaponin A3 Inhibits Palmitate and Stearate through Suppression of SREBP-1 in Pancreatic cancer
Dear Editors,
Thanks to the reviewers and Pharmaceutics for taking the time to review the article. We respect the opinions of all the judges and have carefully revised and responded to the minor reviews of the judges.
We hope that the revised manuscript will better meet the publication requirements of the journal. We would like to thank the editors and reviewers of Pharmarceutics once again for the progressive review of our paper.
Sincerely,
Hyeung-Jin Jang
Responses to reviewers’ comments
Comments and Suggestions for Authors
The authors Kim et al. have submitted second revised version of the manuscript. While the authors successfully revised introductory section, they could not address the concerns raised by this reviewer in a way of additional experiments (especially about clarification of the role of phosphorylated ACC in the manuscript).
Since the authors keep the data in Figure 1d, “TA3… inhibits SREBP-1 expression” should be changed to “tends to inhibit” or similar, because the data is not significant.
Our response: Thank you. As you pointed out, we modified the sentence to “Figure 1. TA3 affects regulation of enzymes involved in lipid metabolism”.
Molecular weight labeling is still missing for figure 4a, 4c, and 4d.
Our response: Thanks for pointing it out. It seems to have been omitted due to several revisions. We added again.
Line 455: “… independent of Ask…” “Akt” instead of “Ask”?
Our response: Thanks for pointing it out. We corrected it with “Akt”.
